# MODEL AGNOSTIC INTERPRETABILITY FOR MULTIPLE INSTANCE LEARNING

**Joseph Early, Christine Evers & Sarvapali Ramchurn**
Agents, Interaction and Complexity Group
Department of Electronics and Computer Science
University of Southampton
`{J.A.Early, C.Evers, sdr1}@soton.ac.uk`

## ABSTRACT

In Multiple Instance Learning (MIL), models are trained using bags of instances, where only a single label is provided for each bag. A bag label is often only determined by a handful of key instances within a bag, making it difficult to interpret what information a classifier is using to make decisions. In this work, we establish the key requirements for interpreting MIL models. We then go on to develop several model-agnostic approaches that meet these requirements. Our methods are compared against existing inherently interpretable MIL models on several datasets, and achieve an increase in interpretability accuracy of up to 30%. We also examine the ability of the methods to identify interactions between instances and scale to larger datasets, improving their applicability to real-world problems.

## 1 INTRODUCTION

In Multiple Instance Learning (MIL), data is organised into bags of instances, and each bag is given a single label. This reduces the burden of labelling, as each instance does not have to be assigned a label, making it useful in applications where labelling is expensive, such as healthcare (Carbonneau et al., 2018). However, there are often only a few key instances within a bag that determine the bag label, so it is necessary to identify these key instances in order to interpret a model's decision-making (Liu et al., 2012). Directly interpreting the decision-making process of machine learning methods is difficult due to the complexity of the models and the scale of the data on which they trained, so there is a need for methods that allow insights into the decision-making processes (Gilpin et al., 2018).

In MIL problems, only some of the instances in each bag will be discriminatory, i.e., a significant number of the instances in a bag will be non-discriminatory or even related to other bag classes (Amores, 2013). Identifying the important instances and presenting those as the interpretation of model decision-making filters out the non-discriminatory information and provides the explainee with a reduced and relevant interpretation, which will avoid overloading the user (Li et al., 2015). Our work provides the following novel contributions:

1. **MIL interpretability definition** We identify two questions that interpretability methods for MIL should be able to answer: 1) Which are the key instances for a bag? 2) What outcome (class/value) does each key instance support? In the rest of this work, we will refer to these two questions as the *which* and the *what* questions respectively. As we explore in Section 2, existing interpretability methods are able to answer *which* questions with varying degrees of accuracy, but can only answer *what* questions under certain assumptions.

2. **MIL model-agnostic interpretability methods** Existing interpretability methods are often model-specific, i.e., they can only be applied to certain types of MIL models. To this end, we build upon the state-of-the-art MIL inherent interpretability methods by developing model-agnostic interpretability methods that are up to 30% more more accurate. The model-agnostic interpretability methods that we propose can be applied to any MIL model, and are able to answer both *which* and *what* questions.

3. **MIL interpretability method comparison** We compare existing model-specific methods with our model-agnostic methods on several MIL datasets. Our experiments are also carried out on four types of MIL model, giving a comprehensive comparison of interpretability performance. To the best of the authors' knowledge, this is one of the first studies to compare different interpretability methods for MIL.[1]

---

[1]Source code for this project is available at `https://github.com/JAEarly/MILLI`.

The remainder of this work is laid out as follows. Section 2 provides relevant background knowledge and reviews existing MIL interpretability methods. Section 3 outlines the requirements for MIL interpretability and details our approaches that meet these requirements. Next, Section 4 provides our results and experiments. Section 5 discusses our findings, and Section 6 concludes.

## 2 BACKGROUND AND RELATED WORK

The standard MIL assumption (SMIL) is a binary problem with positive and negative bags (Dietterich et al., 1997; Maron & Lozano-Pérez, 1998). A bag is positive if any of its instances are positive, otherwise it is negative. The assumptions of SMIL can be relaxed to allow more generalised versions of MIL, e.g., through extensions that include additional positive classes (Scott et al., 2005; Weidmann et al., 2003). SMIL also assumes there is no interaction between instances. However, recent work has highlighted that modelling relationships between instances is beneficial for performance (Tu et al., 2019; Zhou et al., 2009). Existing methods for interpreting MIL models often rely on the SMIL assumption, so cannot generalise to other MIL problems. In addition, existing methods are often model-specific, i.e., they only work for certain types of MIL models, which constrains the choice of model (Ribeiro et al., 2016a). Identifying the key instances in MIL bags is a form of local interpretability (Molnar, 2020), as the key instances are detected for a particular input to the model. Two of the key motivators for interpreting the decision-making process of a MIL system are reliability and trust — identifying the key instances allows an evaluation of the reliability of the system, which increases trust as the decision-making process becomes more transparent. In this work, we use the term interpretability rather than explainability to convey that the analysis remains tied to the models, i.e., these methods do not provide non-technical explanations in human terms.

Under the SMIL assumption, *which* and *what* questions are equivalent — there are only two classes (positive and negative), and the only key instances are the instances that are positive, therefore once the key instances are identified, it is also known what outcome they support. However, when there are multiple positive classes, if instances from different classes co-occur in the same bags, answering *which* and *what* questions becomes two distinct problems. For example, some key instances will support one positive class, and some key instances will support another. Therefore, solely identifying *which* are the key instances does not answer the second question of *what* class they support. Existing methods, such as key instance detection (Liu et al., 2012), MIL attention (Ilse et al., 2018), and MIL graph neural networks (GNNs; Tu et al. (2019)) do not condition their output on a particular class, so it is not apparent what class each instance supports, i.e., they can only answer *which* questions. One existing method that can answer *what* questions is mi-Net (Wang et al., 2018), as it produces instance-level predictions as part of its processing. However, these instance-level predictions do not take account of interactions between the instances, so are often inaccurate. A related piece of work on MIL interpretability is Tibo et al. (2020), which considers interpretability within the scope of multi-multi-instance learning (MMIL; Tibo et al. (2017); Fuster et al. (2021)). In MMIL, the instances within a bag are arranged into into further bags, giving a hierarchical bags-of-bags structure. The interpretability techniques presented by Tibo et al. (2020) are model-specific as they are only designed for MMIL networks. In this work, we aim to overcome the limitations of existing methods by developing model-agnostic methods that can answer both *which* and *what* questions.

In single instance supervised learning, model-agnostic techniques have been developed to interpret models. Post-hoc local interpretability methods, such as Local Interpretable Model-agnostic Explanations (LIME; Ribeiro et al. (2016b)) and SHapley Additive exPlanations (SHAP; Lundberg & Lee (2017)), work by approximating the original predictive model with a locally faithful surrogate model that is inherently interpretable. The surrogate model learns from simplified inputs that represent perturbations of the original input that is being analysed. In this work, one of our proposed methods is a MIL-specific version of this approach.

## 3 METHODOLOGY

At the start of this section, we outline the requirements for MIL interpretability (Section 3.1). In Section 3.2, we propose three methods that meet these requirements under the assumption that there are no interactions between instances (independent-instance methods). In Section 3.3 we remove this assumption and propose our local surrogate model-agnostic interpretability method for MIL.

### 3.1 MIL INTERPRETABILITY REQUIREMENTS

A general MIL classification problem has $C$ possible classes, with one class being negative and the rest being positive. This is a generalisation of the SMIL assumption, which is a special case when $C = 2$. An interpretability method that can only provide the general importance of an instance (without associating it with a particular class) can only answer *which* questions. In order to answer *what* questions, the method needs to state which classes each instance supports and refutes. Formally, for a bag of instances $X = \{x_1, \ldots, x_k\}$ and a bag classification function $F$, we want to assign a value to each instance that represents whether it supports or refutes a class $c$:

$$\mathcal{I}(X, F, c) = \{\phi_1, \ldots, \phi_k\}$$

where $\mathcal{I}$ is the interpretability function and $\phi_i \in \mathbb{R}$ is the interpretability value for instance $i$ with respect to class $c$. Here, we assume that $\phi_i > 0$ means instance $i$ supports class $c$, $\phi_i < 0$ implies instance $i$ refutes class $c$, and the greater $|\phi_i|$, the greater the importance of instance $i$. For some existing methods, such as attention, $\phi_i$ is the same for all classes, and $\phi_i \geq 0$ in all cases, so these requirements are not met. We later demonstrate these limitations in Section 5. In the next two sections we propose several model-agnostic methods that satisfy these requirements.

### 3.2 DETERMINING INSTANCE ATTRIBUTIONS

In this section, we propose three model-agnostic methods for interpreting MIL models under the assumption that the instances are independent. This means we can observe the effects of each instance in isolation without worrying about interactions between the instances. We exploit a property of MIL models: the ability to deal with different sized bags. As MIL models are able to process bags of different sizes, it is possible to remove instances from the bags and observe any changes in prediction, allowing us to understand what instances are responsible for the model's prediction. Below, we propose three methods that use this property to interpret a model's decision making.

**Single** Given a bag of instances $X = \{x_1, \ldots, x_k\}$ and a bag classification function $F$, we can take each instance in turn and form a single instance bag: $X_i = \{x_i\}$ for $i \in \{1, \ldots, k\}$. We then observe the model's prediction on each single instance bag $\phi_i = F_c(X_i)$, where $F_c$ is the output of $F$ for class $c$ (i.e., the $c^{th}$ entry in the output vector of $F(X_i)$). A large value for $\phi_i$ suggests instance $x_i$ supports $c$, and a value close to zero suggest it has no effect with respect to class $c$. If we repeat this over all instances and all classes, we can build a picture of the classes that each instance supports, allowing us to answer *what* questions. However, this method cannot refute classes (i.e., $\phi_i \geq 0$ in all cases). It should be noted that this method gives the same outputs as the inherent interpretability of mi-Net (Wang et al., 2018), but here it is a model-agnostic rather than a model-specific method.

**One Removed** A natural counterpart to the Single method is the One Removed method, where each instance is removed from the complete bag in turn, i.e., we form bags $X_i = X \setminus \{x_i\}$. For a particular class $c$, we can then observe the change in the model's prediction caused by removing $x_i$ from the bag: $\phi_i = F_c(X) - F_c(X_i)$. If the prediction decreases, $x_i$ supports $c$, and if it increases, $x_i$ refutes $c$, i.e., this method is able to both support and refute different classes. However, if there are other instances in the bag that support or refute class $c$, we may not observe a change in prediction when $x_i$ is removed, even if $x_i$ is a key instance.

**Combined** In order to access the benefits of both the Single and One Removed methods, we can combine their outputs. A simple approach is to take the mean, i.e., $\phi_i = \frac{1}{2}[F_c(\{x_i\}) + F_c(X) - F_c(X \setminus \{x_i\})]$. This method can identify the important instances revealed by the Single method, and also refute outcomes as revealed by the One Removed method.

With these three methods, it is assumed that there are no interactions between the instances. This is not true for all datasets, therefore, in the next section, we remove this assumption and propose a further method that is able to deal with the interactions between instances.

### 3.3 DEALING WITH INSTANCE INTERACTIONS

In order to calculate instance attributions whilst accounting for interactions between instances, we have to consider the effect of each instance within the context of the bag. With instance interactions,

the co-occurrence of two (or more) instances changes the bag label from what it would be if the two instances were observed independently. The instances have different meanings depending on the context of the bag, i.e., on their own they mean something different to what they mean when observed together. One way to uncover these instance interactions is to perturb the original input to the model and observe the outcome. In the case of MIL, these perturbations can take the form of removing instances from the original bag. By sampling coalitions of instances, and fitting a weighted linear regression model against the coalitions and their respective model predictions, it is possible to construct a surrogate locally faithful interpretable model that accounts for the instance interactions in the original bag. A coalition is a binary vector $z \in \{0,1\}^k$ that represents a subset $S = \{x_i | z_i = 1\}$, so the number of ones in the coalition $|z|$ is equal to the length of the $S$. The surrogate model $g_c$ takes the form

$$g_c(z) = \phi_0 + \sum_{i=1}^{k} \phi_i z_i, \tag{1}$$

where each coefficient $\phi_i \in \{\phi_1, \ldots, \phi_k\}$ is the importance attribution for each instance $x_i \in X$ with respect to class $c$. Given a collection of $n$ coalitions $Z$, minimising the loss function

$$L(F_c, g_c, \pi) = \sum_{z \in Z} [F_c(S) - g_c(z)]^2 \pi(z), \tag{2}$$

means $g_c$ is a locally faithful approximation of the original model $F$ for class $c$. The loss function is weighted by a kernel $\pi$, which determines how important it is for $g_c$ to be faithful for each individual coalition. Here, $g_c$ only approximates $F$ for class $c$, i.e., to produce interpretations for a bag with respect to all classes, a surrogate model needs to be fit for each class. Similar approaches have been applied in single instance supervised learning in methods such as LIME and KernelSHAP. Both use different choices for $\pi$: LIME employs $l_2$ or cosine distance, and KernelSHAP uses a weighting scheme that approximates Shapley values (Shapley, 1953). For single instance supervised learning models, when perturbing the inputs, it is not possible to simply 'remove' a feature from an input as the models expect fixed-size inputs — either a new model has to be re-trained without that particular feature, or appropriate sampling from other data has to be undertaken, which can lead to unrealistic synthetic data. In MIL, as models are able to deal with different size bags, instances can be removed from the bags without the need for re-training or sampling from other data, meaning these issues do not occur in our setting.

While it is possible utilise the weight kernels from LIME and KernelSHAP for MIL, we identify a significant drawback with both methods. Their choice for $\pi$ weights all coalitions of the same size equally, i.e., they do not consider the content of the coalitions, only their size. Just because two coalitions are of the same size does not mean it is equally important that the surrogate model is faithful to both of them. Furthermore, the sampling strategies of both approaches lead to very large ($|z|$ close to 1) or very small coalitions ($|z|$ close to 0). Unless the number of samples $n$ is very large, samples of average size ($|z|$ close to 0.5) will not be chosen. As we see in Section 4.4, this approach to sampling is appropriate for some datasets, but not for others.

We aim to overcome both of these drawbacks with our own MIL-specific choice of weight kernel and sampling approach. Below, we propose Multiple Instance Learning Local Interpretations (MILLI). At the core of our approach is a new method for weighting coalitions based on an initial ranking of instance importances $r_i \in \{r_1, \ldots, r_k\}$. For a coalition $z$ and ranking of instance importances $r$, we define our weight kernel $\pi_M$ as:

$$\pi_M(z, r) = \frac{1}{|z|} \sum_{i=0}^{k} z_i \, \pi_R(r_i), \tag{3}$$

$$\text{where } \pi_R(r_i) = \begin{cases} (2\alpha - 1)(1 - \frac{r_i}{k})e^{-\hat{\beta} r_i} + 1 - \alpha, & \text{if } \hat{\beta} \geq 0, \\ (1 - 2\alpha)(1 + \frac{r_i - k}{k})e^{|\hat{\beta}|(r_i - k)} + \alpha, & \text{otherwise,} \end{cases} \tag{4}$$

$$\text{and } \hat{\beta} = \begin{cases} \beta, & \text{if } \alpha < 0.5, \\ -\beta, & \text{otherwise.} \end{cases}$$

$\pi_R$ is a function that weights instances based on their order in the ranking. Its two hyperparameters, $\alpha \in [0, 1]$ and $\beta \in (-\infty, \infty)$ define the shape of the function, and ultimately determine the kernel

$\pi_M$. The value of $\alpha$ dictates whether the sampling should be biased towards instances that are highly ranked or not: $\alpha > 0.5$ means $\pi_R$ is biased towards instances higher in ordering, and $\alpha < 0.5$ means $\pi_R$ is biased towards values lower in the ordering. The value of $\beta$ is discussed below, as it becomes important when sampling coalitions. We provide an illustration of $\pi_R$ in Figure 1.

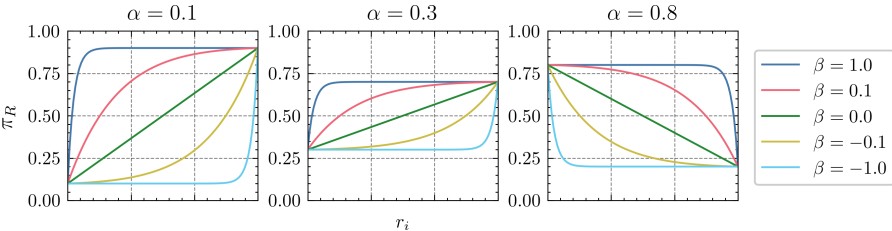

Figure 1: The effect of $\alpha$ and $\beta$ on $\pi_R$ (Equation 4).

In $\pi_M$, the coalition $z$ is weighted on its content rather than its length, i.e., the distance between a subset and a bag is no longer determined simply by the number of instances removed. This is beneficial, as removing many non-discriminatory instances from the bag likely means the label of the bag remains the same, despite being different in size. Conversely, removing one discriminatory instance could change the bag label, even though the bag is only one instance smaller. As well as using $\pi_R$ in the weight kernel, we also utilise it to sample coalitions. We can consider the problem of sampling as a repeated coin toss, where $P(z_i = 1) = p_i$ and $P(z_i = 0) = 1 - p_i$. In equal random sampling, every value $p_i \in \{p_1, \ldots, p_k\}$ is equal to 0.5, meaning $\mathbb{E}[|z|] = 0.5k$. However, it is possible to improve upon equal random sampling by changing the value of $p$ for each instance in bag, i.e., we can change the likelihood of each instance being involved in a coalition. To sample more informative coalitions, we set $p_i = \pi_R(r_i)$, meaning $\mathbb{E}[|z|] = \int_0^k \pi_R \, dr$:

$$\mathbb{E}[|z|] = \begin{cases} \frac{2\alpha-1}{k\hat{\beta}^2}(e^{-\hat{\beta}k} + \hat{\beta}k - 1) + k(1-\alpha), & \text{if } \hat{\beta} \geq 0, \\ \frac{1-2\alpha}{k\hat{\beta}^2}(e^{-\hat{\beta}k} + \hat{\beta}k - 1) + k\alpha, & \text{otherwise.} \end{cases} \tag{5}$$

If $\beta < 0$, the sampling is biased towards smaller coalitions, and if $\beta > 0$, the sampling is biased towards larger coalitions. The maximum and minimum $\mathbb{E}[|z|]$ is controlled by $\alpha$. When $\alpha = 0.5$ or $\beta = 0$, every value $p_i \in \{p_1, \ldots, p_k\}$ is equal to 0.5, meaning we have equal random sampling as described above. We provide an illustration of how $\mathbb{E}[|z|]$ changes in Figure 2.

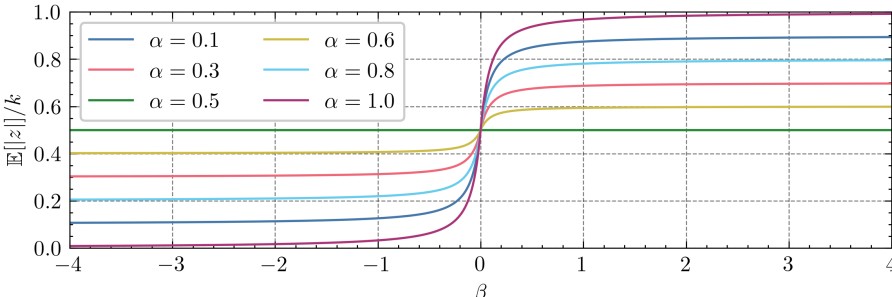

Figure 2: The effect of $\alpha$ and $\beta$ on $\mathbb{E}[|z|]$ (Equation 4). We give the expected coalition size as a proportion of the bag size, i.e., $\mathbb{E}[|z|]/k$.

The final part of MILLI is how to determine the initial ranking of importances. For this, we can use the Single method from Section 3.2, which produces values $\{\phi_1, \ldots, \phi_k\}$, and we convert this to an instance ranking: the new values $\{r_1, \ldots, r_k\}$ represent the position for instance $i$ in $\{\phi_1, \ldots, \phi_k\}$ (e.g., the instance with the greatest value for $\phi$ has $r = 0$). MILLI is more expensive to compute that the methods outlined in Section 3.2: $\mathcal{O}(Cnk^2)$ compared with $\mathcal{O}(Ck)$. However, when there are interactions between the instances, this extra complexity is required in order to build a better understanding of the classes that each instance supports and refutes, allowing more accurate answering of *what* questions. It is important to note that MILLI is indeed model agnostic — it only requires access to the bag classification function $F$, and makes no assumptions about the underlying MIL model. As we demonstrate in the next section, this means we can apply it to any MIL model.

## 4 EXPERIMENTS

We apply our model-agnostic methods to seven MIL datasets. In this section, we detail the evaluation strategy (Section 4.1), the datasets (Section 4.2), models (Section 4.3), and results (Section 4.4). For implementation details, see Appendix A.1.

### 4.1 EVALUATION STRATEGY

For our evaluation, we do not assume that the interpretability methods have consistent output domains; we only assume a larger value implies larger support. Therefore, the best approach for evaluating the interpretability methods is to use ranking metrics — rather than looking at the absolute values produced by the methods, we compare the relative orderings they produce. As noted by Carbonneau et al. (2018), although a large number of benchmark MIL datasets exist, many do not have instance labels. Without instance labels, evaluating interpretability is challenging, as we do not have a ground truth instance ordering to compare to. We identify two appropriate evaluation metrics: for datasets with instance labels, we propose the use of normalised discounted cumulative gain at n (NDCG@n), and for datasets without instance labels, we propose the use of area under the perturbation curve compared with random orderings (AOPC-R; Samek et al. (2016)). While AOPC-R does not require instance labels, it is very expensive to compute. A significant difference between the two metrics is the way they weight the importance ordering: NDCG@n prioritises performance at the start of the ordering, whereas AOPC-R equally prioritises performance across the entire ordering. We are the first to propose both of these metrics for use in evaluating MIL interpretability, and further discuss the advantages of both metrics in Appendix A.3.

### 4.2 DATASETS

Below we detail the three main datasets on which we can evaluate our interpretability methods. These datasets were selected as they have instance labels, so we can evaluate them using both NDCG@n and AOPC-R. However, we provide further results on the classical MIL datasets Musk, Tiger, Elephant and Fox in Appendix A.4 (Dietterich et al., 1997; Andrews et al., 2002).

**SIVAL** The Spatially Independent, Variable Area, and Lighting (SIVAL; Rahmani et al. (2005)) dataset consists of 25 classes of complex objects photographed in different environments, where each class contains 60 images. Each image has been segmented into approximately 30 segments, and each segment is represented by a 30-dimensional feature vector that encodes information such as the segment's colour and texture. The segments are labelled as containing the object or containing background. We selected 12 of the 25 classes to be positive classes, and randomly sampled from the other 13 classes to form the negative class. For additional details see Appendix A.6.

**Four-class MNIST-Bags** The SIVAL dataset has no co-occurrence of instances from different classes, i.e., each bag only contains one class of positive instances. Therefore, the *which* and *what* questions are the same. In order to explore what happens when there are different classes of positive instances in the same bag, we propose an extension of the MNIST-Bags dataset introduced by Ilse et al. (2018). Our extension, four-class MNIST-Bags (4-MNIST-Bags) is setup as follows: class 1 if **8** in bag, class 2 if **9** in bag, class 3 in **8** and **9** in bag, and class 0 otherwise.[2] In this dataset, answering *what* questions goes beyond answering *which* questions, e.g., the existence of an **8** supports classes one and three, but refutes classes zero and two. For further details see Appendix A.7.

**Colon Cancer** To test the applicability of the interpretability methods on larger bag sizes, we apply it to colorectal cancer tissue classification. The ColoRectal Cancer (CRC) dataset (Sirinukunwattana et al., 2016) is a collection of microscopy images with annotated nuclei. We follow the same setup as Ilse et al. (2018), in which a bag is positive if it contains one or more nuclei from the epithelial class. This means the problem conforms to the SMIL assumption, and there are no interactions between instances. Each microscopy image is 500 x 500 pixels, and was split into 27 x 27 pixel patches to give a maximum of 324 patches per slide (patches were discarded if they contained mostly slide background). Not all of the instances are labelled, so we tailor our assessment using NDCG@n to only consider labelled instances. For additional details see Appendix A.8.

---

[2]We use **n** to refer to an image from the MNIST dataset that represents the number $n$, even though that is not the assigned class label in the 4-MNIST-Bags dataset.

### 4.3 MODELS AND METHODS

To highlight the model-agnostic abilities of the proposed methods, for each dataset, we trained four different types of multi-class MIL model: an embedding-based multiple instance neural network (MI-Net; Wang et al. (2018)), an instance-based multiple instance network (mi-Net; Wang et al. (2018)), an attention-based model (MI-Attn; Ilse et al. (2018)), and a graph neural network model (MI-GNN; Tu et al. (2019)). Aside from MI-Net, each of these models provide their own inherent interpretability method that we can compare our methods against. Additional details on the models are given in Appendix A.2. As well as evaluating our independent-instance methods and MILLI, we also compare against LIME and SHAP using both random and guided sampling (choosing coalitions that maximise the weight kernel). This means that in total we are evaluating nine different interpretability methods: inherent interpretability, the three independent-instance methods (Section 3.2), two LIME methods, two SHAP methods, and MILLI (Section 3.3). Note that, aside from the inherent interpretability methods, these methods are either novel (independent-instance methods and MILLI) or are applied to MIL for the time in this study (LIME and SHAP).

### 4.4 RESULTS

For each dataset, we measured the performance of each interpretability method on each of the four MIL models. For the SIVAL dataset we measured the performance on only the negative class and the bag's true class, but for all the other datasets we evaluated the interpretability over every class. This distinction was made as we know that each SIVAL bag can only contain instances from one positive class, therefore it is not necessary to evaluate over all possible classes. We present the interpretability results run against the test set for the SIVAL, 4-MNIST-Bags, and CRC datasets in Tables 1, 2 and 3 respectively. The results are averaged over ten repeat trainings of each model, and we also give the test accuracy of each of the underlying MIL models. We discuss our choice of hyperparameters in Appendix A.5.

By analysing the NDCG@n interpretability results across these three datasets, we find that MILLI performs best with an average of 0.85, followed by the GuidedSHAP and Combined methods (both with an average of 0.81). For the average AOPC-R results (including the results on the classical MIL datasets, see Appendix A.4), we find that Combined, GuidedSHAP, RandomLIME, and MILLI are the best performing methods, however the overall the difference in performance between the methods is much less than what we observe for the NDCG@n metric. For the 4-MNIST-Bags dataset, the difference in performance of MILLI on NDCG@n vs AOPC-R is due to the difference in weighting between the metrics: MILLI achieves a better ordering for the most important instances (outperforms other methods on NDCG@n), but gives the same ordering as other methods for less important instances (equal performance on AOPC-R). In the majority of cases, all of our proposed model-agnostic methods outperform the inherent interpretability methods, and are relatively consistent in performance across all models. For the SIVAL dataset, the independent-instance methods perform well, which is expected as the instances are independent. However, on the 4-MNIST-Bags

Table 1: SIVAL interpretability NDCG@n / AOPC-R results. For all of the interpretability methods, the standard error of the mean was 0.01 or less.

| Methods | MI-Net | mi-Net | MI-Attn | MI-GNN | Overall |
|---|---|---|---|---|---|
| Model Acc | 0.819 | 0.808 | 0.813 | 0.781 | 0.805 |
| Inherent | N/A | 0.813 / 0.265 | 0.717 / 0.005 | 0.586 / 0.023 | 0.705 / 0.098 |
| Single | 0.825 / 0.280 | 0.813 / 0.266 | 0.801 / 0.302 | 0.734 / 0.194 | 0.793 / 0.261 |
| One Removed | 0.778 / 0.256 | **0.837** / 0.308 | 0.736 / 0.293 | 0.776 / 0.231 | 0.782 / 0.272 |
| Combined | **0.828** / 0.291 | 0.828 / 0.294 | **0.803** / 0.316 | 0.762 / 0.227 | 0.805 / 0.282 |
| RandomSHAP | 0.801 / 0.284 | 0.807 / 0.300 | 0.766 / 0.322 | 0.784 / 0.250 | 0.789 / 0.289 |
| GuidedSHAP | 0.826 / 0.291 | 0.819 / 0.290 | 0.790 / 0.313 | 0.765 / 0.235 | 0.800 / 0.282 |
| RandomLIME | 0.809 / **0.295** | 0.815 / **0.310** | 0.776 / **0.335** | **0.793** / **0.258** | 0.798 / **0.299** |
| GuidedLIME | 0.780 / 0.259 | 0.830 / **0.310** | 0.742 / 0.296 | 0.776 / 0.233 | 0.782 / 0.274 |
| MILLI | 0.823 / 0.283 | 0.827 / 0.307 | 0.794 / 0.307 | 0.790 / 0.239 | **0.808** / 0.284 |

Table 2: 4-MNIST-Bags interpretability NDCG@n / AOPC-R results. The MI-GNN model takes four times as long for a single model pass than the other models, so calculating its AOPC-R results on this dataset was infeasible (see Appendix A.3). For all of the interpretability methods, the standard error of the mean was 0.01 or less.

| Methods | MI-Net | mi-Net | MI-Attn | MI-GNN | Overall |
|---|---|---|---|---|---|
| Model Acc | 0.971 | 0.974 | 0.967 | 0.966 | 0.970 |
| Inherent | N/A / N/A | 0.723 / 0.136 | 0.750 / 0.002 | 0.419 / N/A | 0.630 / 0.069 |
| Single | 0.722 / 0.138 | 0.723 / 0.137 | 0.778 / 0.164 | 0.761 / N/A | 0.746 / 0.146 |
| One Removed | 0.811 / 0.187 | 0.810 / 0.186 | 0.809 / 0.143 | 0.786 / N/A | 0.804 / 0.172 |
| Combined | 0.775 / 0.184 | 0.775 / 0.185 | 0.816 / **0.183** | 0.804 / N/A | 0.792 / 0.184 |
| RandomSHAP | 0.813 / 0.186 | 0.809 / 0.185 | 0.825 / 0.178 | 0.828 / N/A | 0.819 / 0.183 |
| GuidedSHAP | 0.773 / 0.187 | 0.773 / 0.188 | 0.816 / **0.183** | 0.805 / N/A | 0.792 / **0.186** |
| RandomLIME | 0.828 / 0.189 | 0.825 / **0.189** | 0.841 / 0.179 | 0.838 / N/A | 0.833 / **0.186** |
| GuidedLIME | 0.760 / 0.189 | 0.756 / 0.187 | 0.785 / 0.145 | 0.776 / N/A | 0.769 / 0.174 |
| MILLI | **0.947** / **0.190** | **0.943** / **0.189** | **0.917** / 0.181 | **0.959** / N/A | **0.942** / **0.186** |

Table 3: CRC interpretability NDCG@n results. As this dataset has much larger bag sizes (264 instances per bag on average), it is infeasible to compute its AOPC-R results (see Appendix A.3). For all of the interpretability methods, the standard error of the mean was 0.02 or less.

| Methods | MI-Net | mi-Net | MI-Attn | MI-GNN | Overall |
|---|---|---|---|---|---|
| Model Acc | 0.795 | 0.795 | 0.830 | 0.770 | 0.797 |
| Inherent | N/A | **0.845** | 0.692 | 0.684 | 0.740 |
| Single | **0.815** | **0.845** | **0.847** | 0.786 | 0.823 |
| One Removed | 0.698 | 0.682 | 0.701 | 0.815 | 0.724 |
| Combined | **0.815** | **0.845** | 0.846 | 0.803 | **0.827** |
| RandomSHAP | 0.695 | 0.690 | 0.717 | 0.703 | 0.701 |
| GuidedSHAP | **0.815** | **0.845** | 0.846 | 0.804 | **0.827** |
| RandomLIME | 0.695 | 0.702 | 0.716 | 0.813 | 0.731 |
| GuidedLIME | 0.687 | 0.699 | 0.701 | **0.818** | 0.726 |
| MILLI | 0.753 | 0.800 | 0.810 | 0.780 | 0.786 |

dataset, MILLI excels as it samples informative coalitions that capture the instance interactions. On the CRC dataset, methods that are able to isolate individual instances, (i.e., the Single, Combined, and GuidedSHAP methods) perform well due to instance independence in this dataset. Furthermore, if we consider the witness rate (WR; the proportion of key instances in each bag; Carbonneau et al. (2018)) of the datasets, we find that the CRC dataset has a higher WR (27.47%) than SIVAL (15.28%) and 4-MNIST-Bags (8.04%). This means, with larger coalitions, it becomes more difficult to isolate the contributions of individual instances, which is why the One Removed and Random sampling methods struggle.

## 5 DISCUSSION

To demonstrate how MILLI captures instance interactions, and to show the limitations of existing methods that cannot condition their output for a particular class, we compare the MILLI interpretations with the attention interpretations on the 4-MNIST-Bags dataset (Figure 3). As this bag contains an **8** and a **9**, the correct label for it is class three. Both the MILLI and attention methods have identified the **8** and **9** instances as key instances, i.e., they have both answered the *which* question. However, only MILLI correctly identifies that the **8** refutes class two and that the **9** refutes class one, answering the *what* question, something that the attention values do not do. We provide further examples, including interpretability outputs for the SIVAL and CRC datasets, in Appendix A.10.

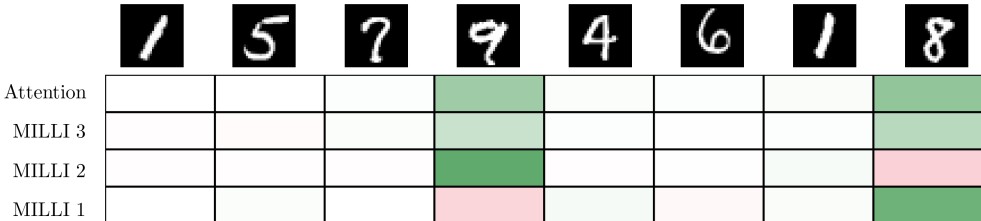

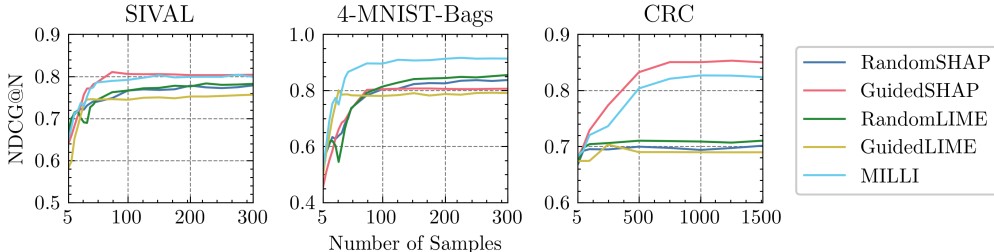

Figure 3: An example of the identification of key instances for the 4-MNIST-Bags experiment. The top row shows the attention value for each instance, and bottom three rows show the output of MILLI for classes three, two, and one respectively (green = support, red = refute).

LIME, SHAP, and MILLI all require the number of sampled coalitions to be selected. A greater number of coalitions means there is more data to fit the surrogate model on, therefore it is more likely to faithful to the underlying model. However, the more samples that are taken, the more expensive the computation. As shown in Figure 4, MILLI is the most consistent in terms of performance and efficiency. For all methods, it is a case of diminishing returns, where additional samples only lead to a small increase in performance.

Figure 4: The effect of sample size on interpretability performance for the MI-Attn model. We provide the results of the same study but for the other MIL models in Appendix A.9.

The different sampling approaches used in this work have distinct advantages. When there are interactions between instances, RandomSHAP and RandomLIME are better than GuidedSHAP and GuidedLIME as they form larger coalitions that are more likely to capture the instance interactions. However, for the CRC dataset, where there are a large number of independent instances and a high WR, RandomSHAP, RandomLIME and GuidedLIME struggle as they cannot form small coalitions. MILLI performs relatively well across all datasets as the size of its sampled coalitions can be adapted depending on the dataset, demonstrating its generalisability. However, it is still outperformed by GuidedSHAP on the CRC dataset. One possible explanation for this is that MILLI is limited to sampling only smaller or larger coalitions, whereas GuidedSHAP samples both large and small coalitions. One approach for improving MILLLI would be to incorporate paired sampling (Covert & Lee, 2020), where for every sampled coalition $z_i$, we also sample its complement coalition $1 - z_i$. Following the advice of Carbonneau et al. (2018), further MIL studies on more complex datasets, such as Pascal VOC (Everingham et al., 2010), could also be insightful for evaluating MIL interpretability methods. It would also be beneficial to examine if these techniques are applicable to MIL domains beyond classification, e.g. MIL regression as in Wang et al. (2020).

## 6 CONCLUSION

In this work, we have discussed the process of model-agnostic interpretability for MIL. Along with defining the requirements for MIL interpretability, we have presented our own approaches and compared them to existing inherently interpretable MIL models. By analysing the methods across several datasets, we have shown that independent-instance methods can be effective, but local surrogate methods are required when there are interactions between the instances. All of our proposed methods are more effective than existing inherently interpretable models, and are able to not only identify *which* are the key instances, but also say *what* classes they support and refute.

REPRODUCIBILITY STATEMENT

For our work, the main details for the methodology are detailed in Section 3. In addition, we provide details that will aid reproducibility in the Appendix. The dataset sources and a general overview of our implementation is given in Appendix A.1. Further information on the MIL models used in this work can be found in Appendix A.2, and specific details on their architectures and training can be found in Appendices A.6, A.7, and A.8 for the SIVAL, 4-MNIST-Bags, and CRC datasets respectively. The codebase for this work can be found on GitHub: https://github.com/JAEarly/MILLI.

ACKNOWLEDGEMENTS

This work was funded by AXA Research Fund and the UKRI Trustworthy Autonomous Systems Hub (EP/V00784X/1). We would also like to thank the University of Southampton and the Alan Turing Institute for their support.

The authors acknowledge the use of the IRIDIS High Performance Computing Facility and associated support services at the University of Southampton in the completion of this work. IRIDIS-5 GPU-enabled compute nodes were used for the long running experiments in this work.

We also acknowledge the use of SciencePlots (Garrett, 2021) for formatting our Matplotlib figures.

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

# A APPENDIX

## A.1 IMPLEMENTATION DETAILS

All code for this work was implemented in Python 3.8, using the PyTorch library for the machine learning functionality. Hyperparameter tuning was carried out using the Optuna libary. Some local experiments were carried out on a Dell XPS Windows laptop, utilising a GeForce GTX 1650 graphics card with 4GB of VRAM. GPU support for machine learning was enabled through CUDA v11.0. Other longer running experiments, such as hyperparameter tuning, were carried out on a remote GPU node utilising a Volta V100 Enterprise Compute GPU with 16GB of VRAM. The longest model to train was the GNN for the CRC dataset, which took just under half an hour. The longest running experiment was the hyperparameter analysis for the CRC dataset at just under 40 hours. This was due to high number of samples for the local surrogate methods and the fact that the results were average over 10 different models. The randomisation of data splits was fixed using seeding; the seed values are provided in the code for each experiment. We use the following sources for our data:

- The annotated SIVAL dataset was downloaded from the publicly accessible page: http://pages.cs.wisc.edu/~bsettles/data/.
- The MNIST dataset was access directly from the PyTorch Python library: https://pytorch.org/vision/stable/datasets.html#mnist.
- The CRC dataset was downloaded from the publicly accessible page: https://warwick.ac.uk/fac/cross_fac/tia/data/crchistolabelednucleihe/.
- The Musk dataset was downloaded from the publicly accessible page: https://archive.ics.uci.edu/ml/datasets/Musk+%28Version+2%29
- The Tiger, Elephant and Fox datasets were downloaded from the publicly accessible page: http://www.cs.columbia.edu/~andrews/mil/datasets.html

## A.2 MODELS

In this work, we trained four different models for each dataset. In this section, we provide further details on each of the models. Each model was tuned independently for each dataset, so in the later sections we provide details of the specific architectures for each dataset. We also tuned the learning rate, weight decay, and dropout for each of the models independently for every dataset. Again, these are given in the later sections for each specific dataset.

**MI-Net** Each instance is embedded to a fixed size, and then these embeddings are aggregated to give a single bag embedding. This aggregation can either take the mean (mil-mean) or max (mil-max) of the instance embeddings. The bag embedding is then classified to give an overall bag prediction. For this model, we tuned the number and size of fully connected (FC) layers, as well as the choice of aggregation function.

**mi-Net** A prediction is made for each individual instance, and then these are aggregated to a single outcome for the bag as a whole. Similar to the MI-Net model, we tuned the FC layers and the aggregation function.

**MI-Attn** Each instance is embedded to a fixed size, and then the mil-attn block produces an attention value for each instance embedding. The instance embeddings are then aggregated to a single bag embedding by performing a weighted sum based on the attention values. The bag embedding is then classified to give an overall bag prediction. The mil-attn block is the same as per the MIL attention mechanism proposed by Ilse et al. (2018), i.e., using a single hidden layer. We tuned the the number and size of the FC layers, as well as the size of the hidden attention layer.

**MI-GNN** The GNN model treats the bag as a fully connected graph and uses graph convolutions to propagate information between instances. Initially, the original instances are embedded to a fixed size. These instance embeddings are then passed onto a $GNN_{embed}$ block and a $GNN_{cluster}$ block, the output of which is used to reduce the graph representation down into a single embedding using differentiable pooling (gnn-pool). The final bag representation is then classified to give an overall bag prediction. For more details see Tu et al. (2019). We tuned the number and size of the embedding, GNN, and classifier layers.

A.3  EVALUATION METRICS

In this section, we provide further details on how we use normalised discounted cumulative gain at n (NDCG@n) to evaluate the interpretability methods for datasets with instance labels, and how we use area under the perturbation curve with random orderings (AOPC-R) to evaluate the interpretability methods for datasets without instance labels.

**NDCG@n**  To use NDCG@n, we first need a ground truth ordering to compare to. For a particular class, we can place each instance into one of three groups based on their ground truth instance labels: supporting instances, neutral instances, and refuting instances. The ideal instance ordering for that class would then have all of the supporting instances at the beginning, followed by all the neutral instances, and then all of the refuting instances at the end. Then, given interpretability outputs $\{\phi_1, \ldots, \phi_k\}$ for a particular class, we rank the outputs from highest to lowest, i.e., in order of how much they support that class. This importance ordering is then compared to the ground truth ordering using the follow metric:

$$\text{NDCG@n} = \frac{1}{\text{IDCG}} \sum_{i=1}^{n} \frac{\text{rel}(i)}{log_2(i+1)},$$

$$\text{where IDCG} = \sum_{i=1}^{n} \frac{1}{log_2(i+1)}.$$

IDCG is the ideal discounted cumulative gain that normalises the scores across different values of $n$. The relevance function $\text{rel}(i)$ is as follows:

$$\text{rel}(i) = \begin{cases} 1 & \text{if the } i^{\text{th}} \text{ instance in the ranking supports the class,} \\ -1 & \text{if the } i^{\text{th}} \text{ instance in the ranking refutes the class,} \\ 0 & \text{otherwise.} \end{cases}$$

**AOPC-R**  Although originally designed for single instance supervised learning, here we adapt AOPC-R for MIL. Given an importance ordering, AOPC successively removes the most relevant instances (i.e., those at the start of the given importance ordering), and measures the change in prediction. A better ordering will show a more rapid decrease in prediction, as instances that are more supportive will be removed first. This rate of decrease in prediction with respect to class $c$ for a classifier $F$ and bag $X = \{x_1, \ldots, x_k\}$ is measured as follows: given the ordered bag $O_X = \{o_1, \ldots, o_k\}$ (which is just $X$ ordered by instance importance, with the most important instances at the start),

$$\text{AOPC} = \frac{1}{k-1} \sum_{i=1}^{k-1} F_c(X) - F_c(X_{MoRF}^{(i)}), \tag{6}$$

$$\text{where } X_{MoRF}^{(0)} = X, \tag{7}$$

$$\text{and } X_{MoRF}^{(i)} = X_{MoRF}^{(i-1)} \setminus \{o_i\}. \tag{8}$$

To normalise the results, one approach is to measure the average difference in AOPC for the given ordering to the AOPC of several random orderings:

$$\text{AOPC-R} = \frac{1}{r} \sum_{i=1}^{r} \text{AOPC}(O_X) - \text{AOPC}(O_{R_i}), \tag{9}$$

where $r$ is the number of random orderings, and $O_{R_i}$ is the $i^{th}$ random ordering of $X$. The repeated calls to $F_c$ make AOPC-R very expensive to compute. This can be reduced by examining the $p$ first perturbations rather than all $k-1$ possible perturbations, but that was not something we investigated in this work (we kept $p = k$ and $r = 10$). Furthermore, due to the use of random orderings, this evaluation metric is inherently stochastic, meaning not only do we have variance in the orderings produced in the methods (e.g., from random sampling), we also have variance due to measurement. NDCG@n does not have either of the issues (i.e., its cheap to compute and deterministic), however it requires (at least some) instance labels.

## A.4 ADDITIONAL RESULTS

In this section, we detail our additional results on the Musk and Tiger, Elephant & Fox (TEF) classical MIL datasets. Although these datasets do not have instance labels, since they are widely used MIL datasets, it is useful to see how our interpretability methods perform on them. Each of these datasets are instance independent, and as they have a low number of instances per bag, we adapted our local surrogate methods to allow sampling with repeats (otherwise we cannot sample enough coalitions). In our previous experiments, we restricted the local surrogate methods to sampling without repeats, which was not an issue with the larger bag sizes in the SIVAL, 4-MNIST-Bags, and CRC datasets. For each of these classic datasets, we used the same model hyperparameters that we used for SIVAL (see Appendix A.6), i.e., we didn't retune the training parameters or model architectures. However, we did tune the parameters for the interpretability methods, which we discuss in Appendix A.5. We give the interpretability results as well as the model performance for Musk (Table A1), Tiger (Table A2), Elephant (Table A3), and Fox (Table A4) below.

Table A1: Musk interpretability AOPC-R results. Note that we are using the Musk1 dataset, rather than Musk2, as the latter has much larger bag sizes, making the use of AOPC-R infeasible.

| Methods | MI-Net | mi-Net | MI-Attn | MI-GNN | Overall |
|---|---|---|---|---|---|
| Model Acc | $0.871 \pm 0.024$ | $0.836 \pm 0.027$ | $0.793 \pm 0.029$ | $0.793 \pm 0.028$ | $0.823 \pm 0.016$ |
| Inherent | N/A | $\mathbf{0.108 \pm 0.010}$ | $0.002 \pm 0.015$ | $0.003 \pm 0.006$ | $0.036 \pm 0.011$ |
| Single | $0.123 \pm 0.010$ | $\mathbf{0.108 \pm 0.010}$ | $0.113 \pm 0.010$ | $\mathbf{0.090 \pm 0.010}$ | $0.108 \pm 0.010$ |
| One Removed | $0.108 \pm 0.009$ | $0.107 \pm 0.010$ | $0.088 \pm 0.008$ | $0.076 \pm 0.009$ | $0.095 \pm 0.009$ |
| Combined | $\mathbf{0.124 \pm 0.010}$ | $0.107 \pm 0.010$ | $\mathbf{0.116 \pm 0.010}$ | $0.088 \pm 0.010$ | $\mathbf{0.109 \pm 0.010}$ |
| RandomSHAP | $0.115 \pm 0.009$ | $0.104 \pm 0.010$ | $0.110 \pm 0.009$ | $0.077 \pm 0.009$ | $0.102 \pm 0.009$ |
| GuidedSHAP | $0.122 \pm 0.009$ | $0.106 \pm 0.010$ | $\mathbf{0.116 \pm 0.010}$ | $0.084 \pm 0.009$ | $0.107 \pm 0.009$ |
| RandomLIME | $0.113 \pm 0.009$ | $0.107 \pm 0.010$ | $0.110 \pm 0.009$ | $0.080 \pm 0.009$ | $0.102 \pm 0.009$ |
| GuidedLIME | $0.117 \pm 0.009$ | $0.106 \pm 0.010$ | $0.102 \pm 0.008$ | $0.077 \pm 0.009$ | $0.101 \pm 0.009$ |
| MILLI | $0.117 \pm 0.009$ | $0.105 \pm 0.010$ | $0.109 \pm 0.009$ | $0.084 \pm 0.010$ | $0.104 \pm 0.009$ |

Table A2: Tiger interpretability AOPC-R results.

| Methods | MI-Net | mi-Net | MI-Attn | MI-GNN | Overall |
|---|---|---|---|---|---|
| Model Acc | $0.827 \pm 0.024$ | $0.807 \pm 0.029$ | $0.807 \pm 0.019$ | $0.800 \pm 0.028$ | $0.810 \pm 0.005$ |
| Inherent | N/A | $0.124 \pm 0.005$ | $0.001 \pm 0.007$ | $0.000 \pm 0.003$ | $0.042 \pm 0.005$ |
| Single | $\mathbf{0.132 \pm 0.005}$ | $0.125 \pm 0.005$ | $0.120 \pm 0.005$ | $\mathbf{0.082 \pm 0.003}$ | $0.115 \pm 0.004$ |
| One Removed | $0.123 \pm 0.005$ | $\mathbf{0.127 \pm 0.005}$ | $0.114 \pm 0.004$ | $0.081 \pm 0.003$ | $0.111 \pm 0.004$ |
| Combined | $0.130 \pm 0.005$ | $\mathbf{0.127 \pm 0.005}$ | $\mathbf{0.124 \pm 0.005}$ | $\mathbf{0.082 \pm 0.003}$ | $\mathbf{0.116 \pm 0.004}$ |
| RandomSHAP | $0.124 \pm 0.004$ | $0.122 \pm 0.005$ | $0.121 \pm 0.005$ | $0.080 \pm 0.003$ | $0.112 \pm 0.004$ |
| GuidedSHAP | $0.130 \pm 0.005$ | $0.125 \pm 0.005$ | $0.121 \pm 0.005$ | $\mathbf{0.082 \pm 0.003}$ | $0.115 \pm 0.004$ |
| RandomLIME | $0.128 \pm 0.005$ | $0.125 \pm 0.005$ | $0.119 \pm 0.005$ | $0.081 \pm 0.003$ | $0.113 \pm 0.004$ |
| GuidedLIME | $0.129 \pm 0.005$ | $0.125 \pm 0.005$ | $0.122 \pm 0.005$ | $\mathbf{0.082 \pm 0.003}$ | $0.115 \pm 0.004$ |
| MILLI | $0.128 \pm 0.005$ | $0.125 \pm 0.005$ | $0.120 \pm 0.004$ | $0.081 \pm 0.003$ | $0.114 \pm 0.004$ |

We find that there is very little difference in interpretability performance for each of our proposed methods across these four datasets. This is to be expected, as the instances are independent, therefore the independent-instance methods as well as the local surrogate methods are able to identify the important instances, i.e., there is little to be gained by sampling coalitions when each instance can be understood in isolation. However, this reinforces the applicability of all of our proposed methods. We note that the attention and GNN inherent interpretability methods perform poorly in these experiments — this is because they are unable to condition their outputs on a specific class (i.e., they can only answer *which* questions, not *what* questions). We also note that the performance is much worse on the Fox dataset. Here, the underlying MIL models perform poorly, so it is unsurprising that the interpretability methods also perform poorly.

Table A3: Elephant interpretability AOPC-R results.

| Methods | MI-Net | mi-Net | MI-Attn | MI-GNN | Overall |
|---|---|---|---|---|---|
| Model Acc | 0.857 ± 0.017 | 0.863 ± 0.017 | 0.867 ± 0.016 | 0.853 ± 0.020 | 0.860 ± 0.003 |
| Inherent | N/A | 0.130 ± 0.006 | 0.000 ± 0.006 | 0.000 ± 0.003 | 0.043 ± 0.005 |
| Single | **0.127 ± 0.004** | **0.131 ± 0.006** | **0.127 ± 0.005** | 0.105 ± 0.004 | **0.122 ± 0.005** |
| One Removed | 0.117 ± 0.004 | 0.129 ± 0.006 | 0.105 ± 0.005 | 0.103 ± 0.004 | 0.114 ± 0.005 |
| Combined | 0.126 ± 0.004 | 0.130 ± 0.006 | **0.127 ± 0.005** | **0.106 ± 0.004** | **0.122 ± 0.005** |
| RandomSHAP | 0.124 ± 0.004 | 0.126 ± 0.006 | 0.119 ± 0.005 | 0.102 ± 0.004 | 0.118 ± 0.005 |
| GuidedSHAP | **0.127 ± 0.004** | 0.130 ± 0.006 | 0.124 ± 0.005 | 0.105 ± 0.004 | **0.122 ± 0.005** |
| RandomLIME | 0.124 ± 0.004 | 0.128 ± 0.006 | 0.121 ± 0.005 | 0.104 ± 0.004 | 0.119 ± 0.005 |
| GuidedLIME | 0.123 ± 0.004 | 0.130 ± 0.006 | 0.118 ± 0.005 | 0.104 ± 0.004 | 0.119 ± 0.005 |
| MILLI | 0.124 ± 0.005 | 0.128 ± 0.006 | 0.121 ± 0.005 | 0.103 ± 0.005 | 0.119 ± 0.005 |

Table A4: Fox interpretability AOPC-R results.

| Methods | MI-Net | mi-Net | MI-Attn | MI-GNN | Overall |
|---|---|---|---|---|---|
| Model Acc | 0.600 ± 0.011 | 0.610 ± 0.017 | 0.620 ± 0.023 | 0.580 ± 0.013 | 0.603 ± 0.007 |
| Inherent | N/A | **0.050 ± 0.002** | 0.001 ± 0.002 | 0.000 ± 0.001 | 0.017 ± 0.002 |
| Single | 0.044 ± 0.002 | 0.049 ± 0.002 | 0.041 ± 0.002 | 0.021 ± 0.001 | 0.039 ± 0.002 |
| One Removed | 0.043 ± 0.002 | 0.049 ± 0.002 | 0.040 ± 0.002 | 0.021 ± 0.001 | 0.038 ± 0.002 |
| Combined | 0.044 ± 0.002 | **0.050 ± 0.002** | 0.041 ± 0.002 | 0.021 ± 0.001 | 0.039 ± 0.002 |
| RandomSHAP | 0.044 ± 0.002 | 0.049 ± 0.002 | 0.041 ± 0.002 | 0.021 ± 0.001 | 0.039 ± 0.002 |
| GuidedSHAP | **0.045 ± 0.002** | **0.050 ± 0.002** | **0.042 ± 0.002** | 0.021 ± 0.001 | **0.040 ± 0.002** |
| RandomLIME | 0.044 ± 0.002 | **0.050 ± 0.002** | 0.041 ± 0.002 | 0.021 ± 0.001 | 0.039 ± 0.002 |
| GuidedLIME | 0.044 ± 0.002 | **0.050 ± 0.002** | 0.041 ± 0.002 | 0.021 ± 0.001 | 0.039 ± 0.002 |
| MILLI | 0.043 ± 0.002 | 0.049 ± 0.002 | 0.041 ± 0.002 | **0.022 ± 0.001** | 0.039 ± 0.002 |

## A.5 INTERPRETABILITY METHOD HYPERPARAMETER SELECTION

In this section we discuss our method for hyperparameter selection in the interpretability methods, and detail the hyperparameters that we found to be most effective. An advantage of the inherent interpretability and independent-instance methods is that they do not have hyperparameters, i.e., we only had to select hyperparameters for the local surrogate interpretability methods. First, we discuss our choice of hyperparameters for LIME, and then our choice of hyperparameters for MILLI.

**LIME hyperparameters** When using the LIME weight kernel, there are two hyperparameters to tune. Firstly, the distance measures that are commonly used are L2 and cosine distance. In our experiments, we found very little difference between each of these distance measures, therefore we arbitrarily chose to use L2 distance in all of our experiments. The second hyperparameter is the kernel width, which determines the weighting of coalitions, i.e., a large kernel width means all coalitions are weighted more evenly, and a small kernel width prioritises larger coalitions. We chose to use a kernel width for all of our experiments that is determined by the average bag size, such that the half coalition (i.e., $|z| = 0.5k$) is weighted at 0.5.

**MILLI hyperparameters** For MILLI, there were three hyperparameters to tune: the number of samples, $\alpha$, and $\beta$. For the number of samples, we generated sample size plots such as Figure 4 (also see Appendix A.9), and chose the number of samples to be at the point where all the methods had reasonably converged. For $\alpha$ and $\beta$ we ran a grid search over the possible values, and chose the best performing pair of parameters. The hyperparameters were tuned for each dataset, except for the TEF datasets, in which we only tuned on Tiger and then used the same hyperparameters across

all three datasets. In Table A5, we provide the chosen hyperparameters for each dataset, as well as the expected coalition size $\mathbb{E}[|z|]$ determined by $\alpha$ and $\beta$. For the CRC, Musk and TEF datasets, the chosen parameters produce small values for $\mathbb{E}[|z|]$, i.e., the sampling is heavily biased towards smaller coalitions, which is expected as these are instance independent datasets. Conversely, the parameters for the 4-MNIST-Bags focus on sampling larger coalitions that are able to capture the instance interactions in the dataset. Although the SIVAL dataset is instance independent, the parameters also focus on sampling larger coalitions. This could be because, although the instances are independent, it may be difficult to classify an object from just one instance, i.e., several instances are needed to make the correct decision. We also note that, in all cases, $\alpha < 0.5$, i.e., the sampling is biased towards instances ranked lower in the initial importance ordering used by MILLI. Our explanation for this is that it is easy to understand the contribution of discriminatory instances, as they have a large effect on the model prediction, but it is more difficult to understand the contribution of non-discriminatory instances, as they have much less of an effect. Therefore, more samples containing non-discriminatory instances are required to properly understand their effect, hence biasing the sampling towards them.

Table A5: MILLI hyperparameters.

| Dataset | Sample Size | $\alpha$ | $\beta$ | $\mathbb{E}[|z|]$ |
|---|---|---|---|---|
| SIVAL | 200 | 0.05 | -0.01 | 13 |
| 4-MNIST-Bags | 150 | 0.05 | 0.01 | 16 |
| CRC | 1000 | 0.008 | -5.0 | 2 |
| MUSK | 150 | 0.3 | -1.0 | 2 |
| TEF | 150 | 0.3 | 0.01 | 3 |

## A.6 SIVAL EXPERIMENT DETAILS

**Dataset** For the SIVAL dataset, each instance is represented by a 30-dimensional feature vector, and there are around 30 instances per bag. We chose 12 of the 25 original classes to be the positive classes, and randomly selected 30 images from each of the other 13 classes to form the negative class, meaning overall we had 13 classes (12 positive and one negative). In total, we had 60 bags for each of the 12 positive classes, and 390 bags for the single negative class, meaning the class distribution was $\approx 5.4\%$ for each positive class and $\approx 35.1\%$ for the negative class. The arbitrarily chosen positive classes were: *apple*, *banana*, *checkeredscarf*, *cokecan*, *dataminingbook*, *goldmedal*, *largespoon*, *rapbook*, *smileyfacedoll*, *spritecan*, *translucentbowl*, and *wd40can*. We normalised each instance according to the dataset mean and standard deviation. No other data augmentation was used. The dataset was separated into train, validation, and test data using an 80/10/10 split. This was done with stratified sampling in order to maintain the same data distribution across all splits.

**Training** When training models against the SIVAL dataset, we used a batch size of one, i.e., a single bag of, on average, 30 instances. We trained the models to minimise cross entropy loss using the Adam optimiser; the hyperparamater details for learning rate (LR), weight decay (WD) and dropout (DO) are given in Table A6. We utilised early stopping based on validation loss — if the validation loss had not decreased for 10 epochs then we terminated the training procedure and reset the model to the point at which it caused the last decrease in validation loss. Otherwise, the maximum number of epochs was 100. The tuned architectures for each model are given in Tables A7 to A10, and the results for each model are comapred in Table A11.

Table A6: SIVAL hyperparameters.

| Model | LR | WD | DO |
|---|---|---|---|
| Mi-Net | $5 \times 10^{-3}$ | $1 \times 10^{-3}$ | 0.45 |
| mi-Net | $5 \times 10^{-4}$ | $1 \times 10^{-5}$ | 0.25 |
| MI-Attn | $1 \times 10^{-3}$ | $1 \times 10^{-5}$ | 0.15 |
| MI-GNN | $5 \times 10^{-4}$ | $1 \times 10^{-5}$ | 0.2 |

Table A7: SIVAL MI-Net architecture.

| Layer | Type | Input | Output |
|---|---|---|---|
| 1 | FC + ReLU + DO | 30 | 128 |
| 2 | FC + ReLU + DO | 128 | 256 |
| 3 | mil-max | 256 | 256 |
| 4 | FC | 256 | 13 |

Table A8: SIVAL mi-Net architecture.

| Layer | Type | Input | Output |
|---|---|---|---|
| 1 | FC + ReLU + DO | 30 | 512 |
| 2 | FC + ReLU + DO | 512 | 256 |
| 3 | FC + ReLU + DO | 256 | 64 |
| 4 | FC | 64 | 13 |
| 5 | mil-mean | 13 | 13 |

Table A9: SIVAL MI-Attn architecture.

| Layer | Type | Input | Output |
|---|---|---|---|
| 1 | FC + ReLU + DO | 30 | 128 |
| 2 | FC + ReLU + DO | 128 | 256 |
| 3 | FC + ReLU + DO | 256 | 128 |
| 4 | mil-attn(256) + DO | 128 | 128 |
| 5 | FC | 128 | 13 |

Table A10: SIVAL MI-GNN architecture.

| Layer | Type | Input | Output |
|---|---|---|---|
| 1 | FC + ReLU + DO | 30 | 128 |
| 2a $GNN_{embed}$ | SAGEConv + ReLU + DO | 128 | 128 |
| | SAGEConv + ReLU + DO | 128 | 256 |
| | SAGEConv + ReLU + DO | 256 | 64 |
| 2b $GNN_{cluster}$ | SAGEConv + Softmax | 128 | 1 |
| 3 | gnn-pool | 64 | 64 |
| 4 | FC + ReLU + DO | 64 | 128 |
| 5 | FC | 128 | 13 |

Table A11: SIVAL model results. The mean performance was calculated over ten repeat trainings of each model, and the standard error of the mean is given.

| Model | Train Accuracy | Val Accuracy | Test Accuracy |
|---|---|---|---|
| MI-Net | **0.984 ± 0.004** | **0.850 ± 0.009** | **0.819 ± 0.012** |
| mi-Net | 0.967 ± 0.005 | 0.835 ± 0.010 | 0.808 ± 0.011 |
| MI-Attn | 0.972 ± 0.007 | 0.830 ± 0.011 | 0.813 ± 0.012 |
| MI-GNN | 0.932 ± 0.014 | 0.803 ± 0.014 | 0.781 ± 0.019 |

### A.7 4-MNIST-BAGS EXPERIMENT DETAILS

**Dataset**  In the 4-MNIST-Bags experiments, the bag sizes were draw from a normal distribution, with a mean of 30 and a variance of 2. We used 2500 training bags, 1000 validation bags, and 1000 test bags. The instances in the training bags were only drawn from the original MNIST training split, and the instances in the validation and test bags were only drawn from the original MNIST test split, i.e., there was no overlap between training, validation, and test instances. The classes were balanced, so, on average, there were 625 bags per class in the training data, and 250 bags per class in the validation and test data. We normalised the MNIST images using the PyTorch normalise transformation, with a mean of 0.1307 and a stand deviation of 0.3081. No other data augmentation was carried out.

**Training**  The training procedure was the same as for the SIVAL experiments: a batch size of one, early stopping with a patience of ten, and a maximum of 100 epochs. The training hyperparamater details are given in Table A12. For each model, we first passed the MNIST instances through a convolutional architecture to produce initial instance embeddings. This encoder was not tuned for each model (i.e., the architecture was fixed, but the weights were learnt). The architecture for this encoder is given in Table A13, and then the model architectures are given in Tables A14 to A17. The encoder produces features vectors with 800 features, therefore the input size to each of the models is 800. The model results are given in Table A18.

Table A12: 4-MNIST-Bags hyperparameters.

| Model | LR | WD | DO |
|---|---|---|---|
| MI-Net | $1 \times 10^{-4}$ | $1 \times 10^{-3}$ | 0.3 |
| mi-Net | $1 \times 10^{-4}$ | $1 \times 10^{-4}$ | 0.3 |
| MI-Attn | $1 \times 10^{-4}$ | $1 \times 10^{-4}$ | 0.15 |
| MI-GNN | $5 \times 10^{-5}$ | $1 \times 10^{-5}$ | 0.3 |

Table A13: 4-MNIST-Bags convolutional encoding architecture. For the convolutional (Conv2d) and pooling (MaxPool2d) layers, the numbers in the brackets are the kernel size, stride, and padding.

| Layer | Type | Input | Out |
|---|---|---|---|
| 1 | Conv2d(5, 1, 0) + ReLU | 1 | 20 |
| 2 | MaxPool2d(2, 2, 0) + DO | 20 | 20 |
| 3 | Conv2d(5, 1, 0) + ReLU | 20 | 50 |
| 4 | MaxPool2d(2, 2, 0) + DO | 50 | 50 |

Table A14: 4-MNIST-Bags MI-Net architecture.

| Layer | Type | Input | Output |
|---|---|---|---|
| 1 | FC + ReLU + DO | 800 | 128 |
| 2 | FC + ReLU + DO | 128 | 512 |
| 3 | mil-mean | 512 | 512 |
| 4 | FC | 512 | 4 |

Table A15: 4-MNIST-Bags mi-Net architecture.

| Layer | Type | Input | Output |
|---|---|---|---|
| 1 | FC + ReLU + DO | 800 | 512 |
| 2 | FC + ReLU + DO | 512 | 128 |
| 3 | FC + ReLU + DO | 128 | 64 |
| 4 | FC | 128 | 4 |
| 5 | mil-mean | 4 | 4 |

Table A16: 4-MNIST-Bags MI-Attn architecture.

| Layer | Type | Input size | Output size |
|---|---|---|---|
| 1 | FC + ReLU + Dropout | 800 | 64 |
| 2 | FC + ReLU + Dropout | 64 | 256 |
| 3 | mil-attn(64) + Dropout | 256 | 256 |
| 4 | FC + ReLU + Dropout | 256 | 64 |
| 5 | FC | 64 | 4 |

Table A17: 4-MNIST-Bags MI-GNN architecture.

| Layer | Type | Input size | Output size |
|---|---|---|---|
| 1 | FC + ReLU + Dropout | 800 | 64 |
| 2 | FC + ReLU + Dropout | 64 | 64 |
| 3a GNN$_{embed}$ | SAGEConv + ReLU + Dropout | 64 | 128 |
| | SAGEConv + ReLU + Dropout | 128 | 128 |
| | SAGEConv + ReLU + Dropout | 128 | 128 |
| 3b GNN$_{cluster}$ | SAGEConv + Softmax | 64 | 1 |
| 4 | gnn-pool | 128 | 128 |
| 5 | FC + ReLU + Dropout | 128 | 64 |
| 6 | FC | 64 | 4 |

Table A18: 4-MNIST-Bags model results. The mean performance was calculated over ten repeat trainings of each model, and the standard error of the mean is given.

| Model | Train Accuracy | Val Accuracy | Test Accuracy |
|---|---|---|---|
| MI-Net | $\mathbf{0.995 \pm 0.001}$ | $0.972 \pm 0.002$ | $0.971 \pm 0.002$ |
| mi-Net | $0.993 \pm 0.001$ | $\mathbf{0.973 \pm 0.002}$ | $\mathbf{0.974 \pm 0.002}$ |
| MI-Attn | $0.991 \pm 0.002$ | $0.967 \pm 0.003$ | $0.967 \pm 0.003$ |
| MI-GNN | $0.984 \pm 0.002$ | $0.968 \pm 0.002$ | $0.966 \pm 0.002$ |

## A.8  CRC EXPERIMENT DETAILS

**Dataset**  In the CRC dataset, there are 100 microscopy images, and each image is annotated with four classes of nuclei: epithelial, inflammatory, fibroblast, and miscellaneous. Of these 100 images, 50 are in the negative class (no epithelial nuclei), and 50 are in the positive class (at least one epithelial nuclei). Each image is 500 x 500 pixels, and was split into 27 x 27 pixel patches to give 324 patches per slide. The patches were created by applying a non-overlapping grid over the image, where each cell of the grid was a 27 x 27 pixel region. The alternative would be to extract patches centred on each marked nuclei (as done by Ilse et al. (2018)), however this method then requires the nuclei to be marked for unseen data. This accounts for the difference between our results and the results of Ilse et al. (2018) — extracting the patches using a fixed grid will include patches that do not contain any marked nuclei, increasing the amount of non-discriminatory data in each bag and thus making the problem harder to learn. We removed slide background patches by using a brightness threshold — any patch with an average pixel value above 230 (using pixel values 0 to 255) was discarded. This left an average of 264 patches per image, and one image was discarded as it had zero foreground patches (this was also manually verified). The images were normalised using the dataset mean (0.8035, 0.6499, 0.8348) and standard deviation (0.0858, 0.1079, 0.0731). During training, we also applied three transformations to patches at random: horizontal flips, vertical flips, and 90 degree rotations. The dataset was separated into train, validation, and test data using a 60/20/20 split. This was done with stratified sampling in order to maintain the same data distribution across all splits.

**Training**  The training procedure was the same as for the SIVAL and 4-MNIST-Bags experiments: a batch size of one, early stopping with a patience of ten, and a maximum of 100 epochs. The training hyperparamater details are given in Table A19. Following the same procedure as the 4-MNIST-Bags experiments, for each model, we first passed the patches through a convolutional architecture to produce initial instance embeddings. The architecture for this encoder is given in Table A20, and then the model architectures are given in Tables A21 to A24. The encoder produces features vectors with 1200 features, therefore the input size to each of the models is 1200. The model results are given in Table A25.

Table A19: CRC training hyperparameters.

| Model | LR | WD | DO |
|---|---|---|---|
| MI-Net | $5 \times 10^{-4}$ | $1 \times 10^{-3}$ | 0.3 |
| mi-Net | $5 \times 10^{-4}$ | $1 \times 10^{-2}$ | 0.25 |
| MI-Attn | $1 \times 10^{-3}$ | $1 \times 10^{-6}$ | 0.2 |
| MI-GNN | $1 \times 10^{-3}$ | $1 \times 10^{-2}$ | 0.35 |

Table A20: CRC convolutional encoding architecture. For the convolutional (Conv2d) and pooling (MaxPool2d) layers, the numbers in the brackets are the kernel size, stride, and padding.

| Layer | Type | Input | Out |
|---|---|---|---|
| 1 | Conv2d(4, 1, 0) + ReLU | 3 | 36 |
| 2 | MaxPool2d(2, 2, 0) + DO | 36 | 36 |
| 3 | Conv2d(3, 1, 0) + ReLU | 36 | 48 |
| 4 | MaxPool2d(2, 2, 0) + DO | 48 | 48 |

Table A21: CRC MI-Net architecture.

| Layer | Type | Input | Output |
|---|---|---|---|
| 1 | FC + ReLU + DO | 1200 | 64 |
| 2 | FC + ReLU + DO | 64 | 512 |
| 3 | mil-max | 512 | 512 |
| 4 | FC + ReLU + DO | 512 | 128 |
| 4 | FC + ReLU + DO | 128 | 64 |
| 4 | FC + ReLU + DO | 64 | 2 |

Table A22: CRC mi-Net architecture.

| Layer | Type | Input | Output |
|---|---|---|---|
| 1 | FC + ReLU + DO | 1200 | 64 |
| 2 | FC + ReLU + DO | 64 | 64 |
| 3 | FC + ReLU + DO | 64 | 64 |
| 4 | FC | 64 | 2 |
| 5 | mil-max | 2 | 2 |

Table A23: CRC MI-Attn architecture.

| Layer | Type | Input size | Output size |
|---|---|---|---|
| 1 | FC + ReLU + Dropout | 1200 | 64 |
| 2 | FC + ReLU + Dropout | 64 | 64 |
| 3 | FC + ReLU + Dropout | 64 | 256 |
| 4 | mil-attn(128) + Dropout | 256 | 256 |
| 5 | FC | 256 | 2 |

Table A24: CRC MI-GNN architecture.

| Layer | Type | Input size | Output size |
|---|---|---|---|
| 1 | FC + ReLU + Dropout | 1200 | 64 |
| 2 | FC + ReLU + Dropout | 64 | 128 |
| 3a GNN$_{embed}$ | SAGEConv + ReLU + Dropout | 128 | 128 |
| 3b GNN$_{cluster}$ | SAGEConv + Softmax | 128 | 1 |
| 4 | gnn-pool | 128 | 128 |
| 5 | FC + ReLU + Dropout | 128 | 128 |
| 6 | FC | 128 | 2 |

Table A25: CRC model results. The mean performance was calculated over ten repeat trainings of each model, and the standard error of the mean is given.

| Model | Train Accuracy | Val Accuracy | Test Accuracy |
|---|---|---|---|
| MI-Net | **0.880 ± 0.041** | 0.805 ± 0.034 | 0.795 ± 0.042 |
| mi-Net | 0.856 ± 0.041 | 0.810 ± 0.043 | 0.795 ± 0.049 |
| MI-Attn | 0.870 ± 0.014 | **0.860 ± 0.017** | **0.830 ± 0.028** |
| MI-GNN | 0.791 ± 0.039 | 0.765 ± 0.031 | 0.770 ± 0.032 |

## A.9 ADDITIONAL EXPERIMENTS

In this section, we provide additional sample size experiments for the SIVAL, 4-MNIST-Bags, and CRC datasets for MI-Net (Figure A1), mi-Net (Figure A2), and MI-GNN (Figure A3). We find the trends are relatively consistent across all the models. GuidedSHAP and MILLI are the most consistent and well performing methods, and MILLI is particularly effective on the 4-MNIST-Bags dataset. One considerable difference is the performance of RandomLIME and GuidedLIME on the CRC dataset for the MI-GNN model; for all other models, both LIME methods perform poorly on the CRC dataset. Further investigation is required to understand why this happens. However, the LIME methods are still outperformed by MILLI and GuidedSHAP on the CRC dataset, even for the MI-GNN model.

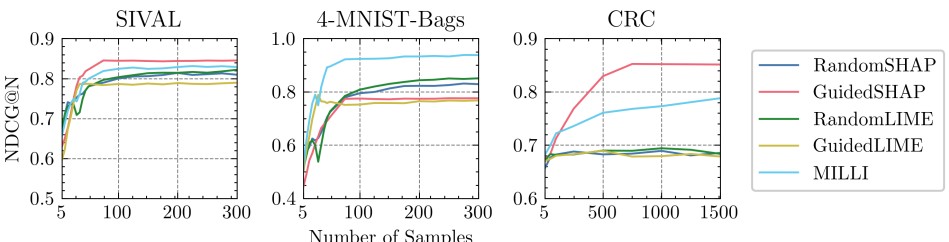

Figure A1: The effect of sample size on interpretability performance for MI-Net.

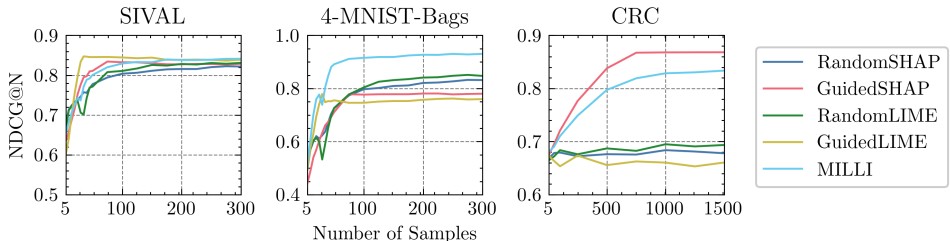

Figure A2: The effect of sample size on interpretability performance for mi-Net.

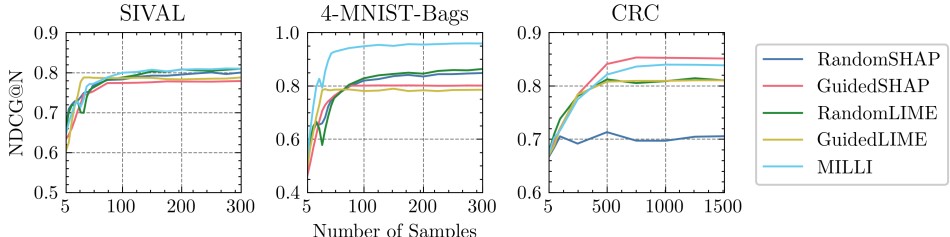

Figure A3: The effect of sample size on interpretability performance for MI-GNN.

## A.10 ADDITIONAL OUTPUTS

In this section, we provide additional interpretability outputs from our studies, similar to the one in Section 5. First, we provide additional outputs for the 4-MNIST-Bags dataset. Then, we show the interpretability outputs for the SIVAL and CRC datasets. The SIVAL outputs are created by ranking the instances in a bag according to some interpretability function (i.e., by using MILLI), and then selecting the top $n$, where $n$ is the known number of key instances in the bag. Then, the corresponding segment for each instance is weighted by the function $log_2(i + 1)^{-1}$, where $i$ is the instance's position in the ranking (this is the same scaling used in NDCG@n). The other instances all received a weighting of zero. The brightness of each segment in the output image then corresponds with its relative importance according to the interpretability method. For the CRC interpretability outputs, the important patches are found by using an interpretability method to output the top $n$ patches that support a particular class, where $n$ is the number of known important instances for that class. We then highlight these patches while dimming the other patches to produce an interpretation of the model's decision-making.

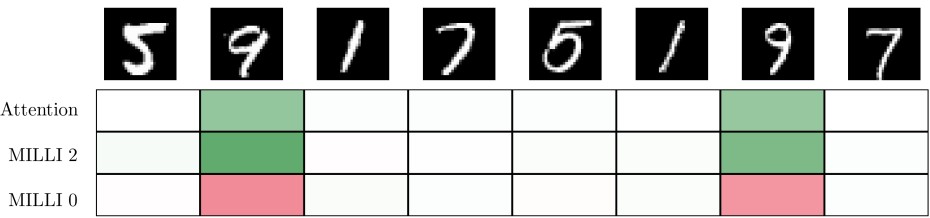

Figure A4: The interpretability output for a class two bag on the 4-MNIST-Bags experiment. The top row shows the attention value for each instance, and bottom two rows show the output of MILLI for classes two and zero respectively (green = support, red = refute). We can see that both Attention and MILLI have identified the **9**s as important, but only MILLI is able to also say that the **9**s support class two and refute class zero.

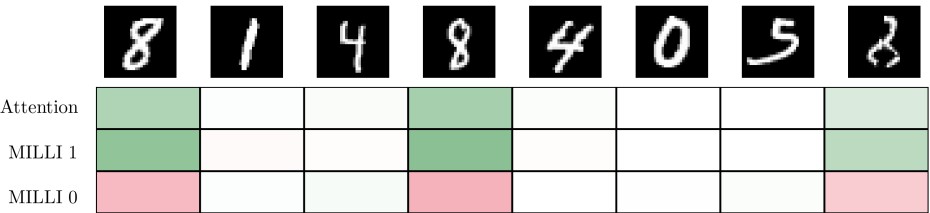

Figure A5: The interpretability output for a class one bag on the 4-MNIST-Bags experiment. We can see that both Attention and MILLI have identified the **8**s as important, but only MILLI is able to also say that the **8**s support class one and refute class zero. Also note that the colours are less strong for the final **8** — the digit is partially obscured, so the model is less confident.

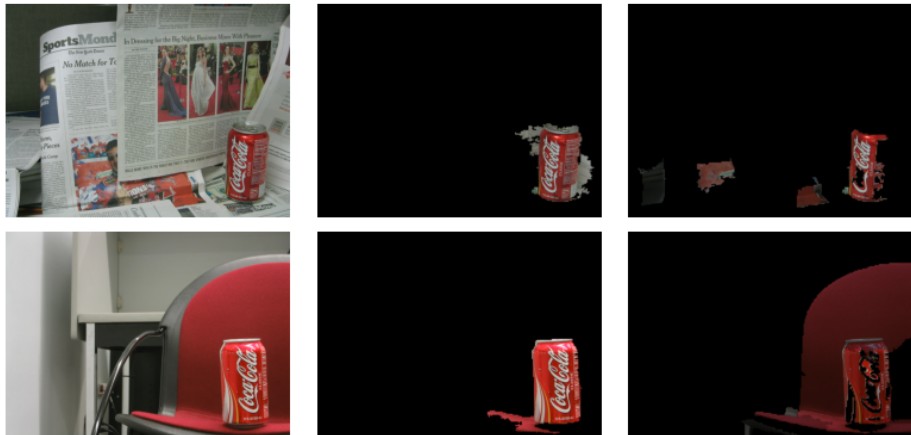

Figure A6: The interpretability output for two examples of the *cokecan* class in the SIVAL dataset. The first column shows the original images, the second column the ground truth segments that contain the object, and the final column is the output from MILLI. The interpretations show the model is relying heavily on the colour red as an indicator of the object's location, as it also picks up on the red in the background as being important.

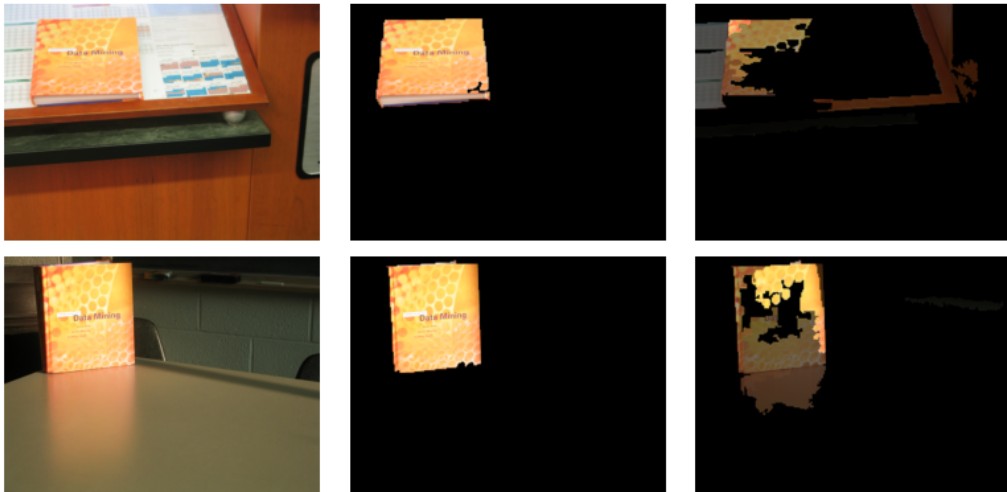

Figure A7: The interpretability output for two examples of the *dataminingbook* class in the SIVAL dataset. Again, the interpretations show the model is relying heavily on the colour orange as an indicator of the object's location, as it also picks up on the similar colours in the background, including the reflection of the book in the table.

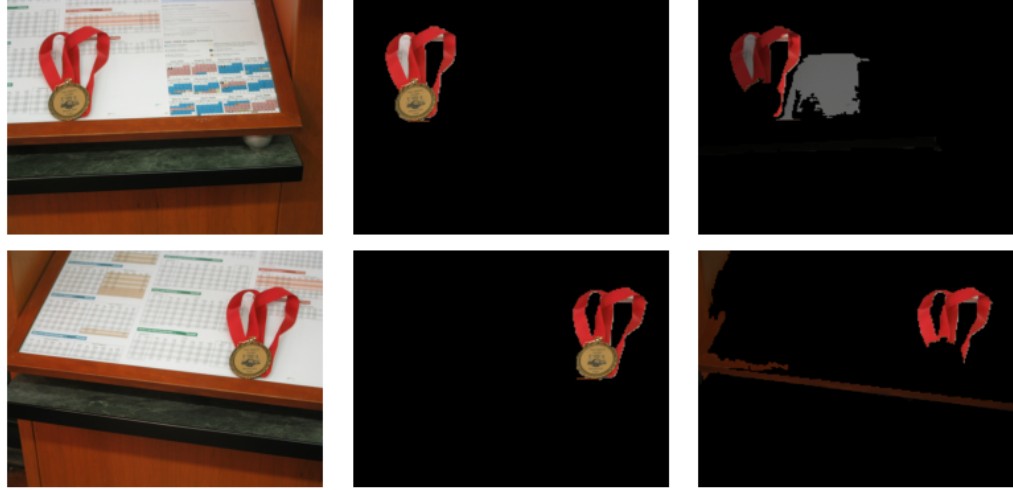

Figure A8: The interpretability output for two examples of the *goldmedal* class in the SIVAL dataset. Here, the interpretations show the model is ignoring the gold medal itself, and instead focusing on the red ribbon, i.e., if it were shown the medal without the ribbon, it would likely misclassify it.

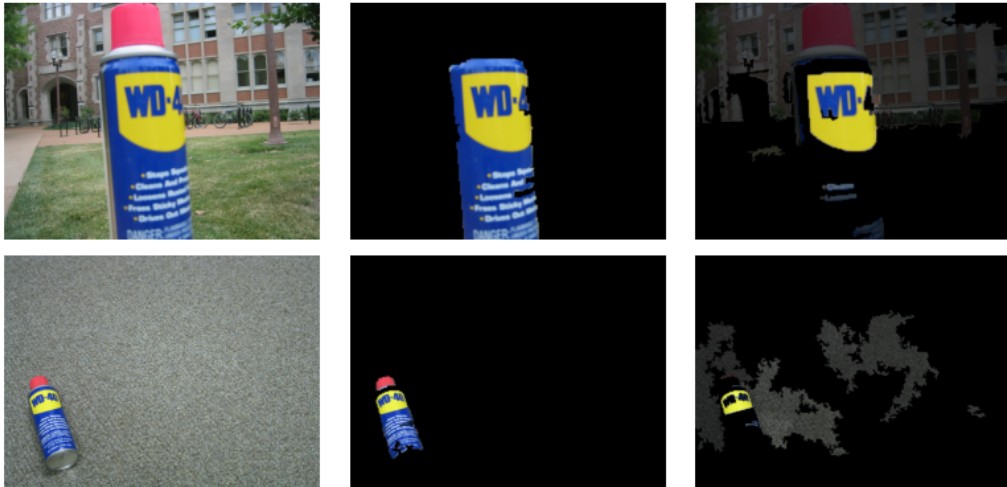

Figure A9: The interpretability output for two examples of the *wd40can* class in the SIVAL dataset. Here, the interpretations show the model is predominantly focusing on the strong yellow colour at the top of the can, and sometimes picks out the letters and red cap. The rest of the bottle is largely ignored, meaning if the top half were obscured, the model would likely misclassify it.

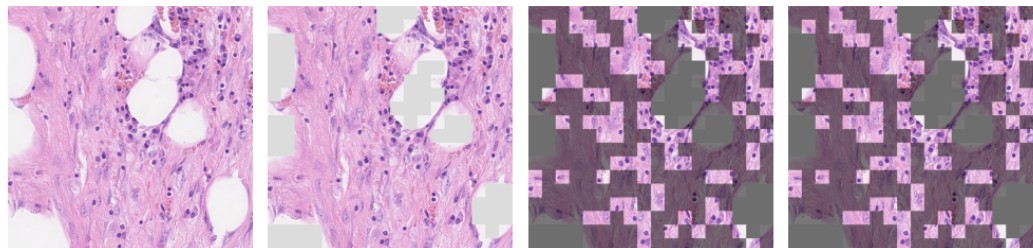

Figure A10: The interpretability output for an image in the CRC dataset. From left to right, the figures shows: the original image, the image with background patches removed, the ground truth patches for the image's label, and the interpretability output from MILLI. In this example, the image is class zero (i.e., non-epithelial), meaning the highlighted patches contain mostly fibroblast and inflammatory nuclei. As shown here, MILLI has identified that the model is using the same patches as the ground truth to make a decision, indicating that the model has learnt to correctly identify fibroblast and inflammatory nuclei as supporting instances for non-epithelial images.

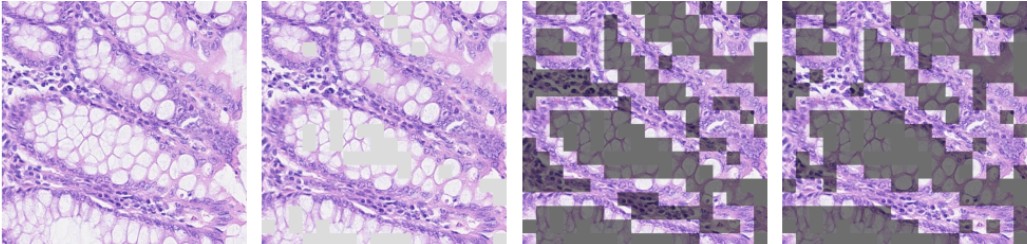

Figure A11: The interpretability output for a positive image in the CRC dataset. In this example, the ground truth patches all contain at least one epithelial nuclei. MILLI has identified that the model is using most of the same patches as the ground truth to make a decision, indicating that the model has learnt to correctly identify epithelial nuclei as supporting class one.

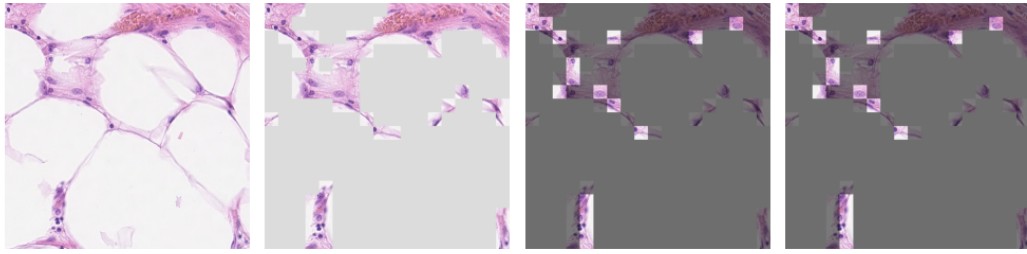

Figure A12: The interpretability output for a negative image in the CRC dataset. In this example, the patches are sparse — there are a lot of background patches that are removed, and the key patches are spread out (as opposed to being connected as in the previous examples). Again, MILLI is able to correctly identify the patches that support the negative class.

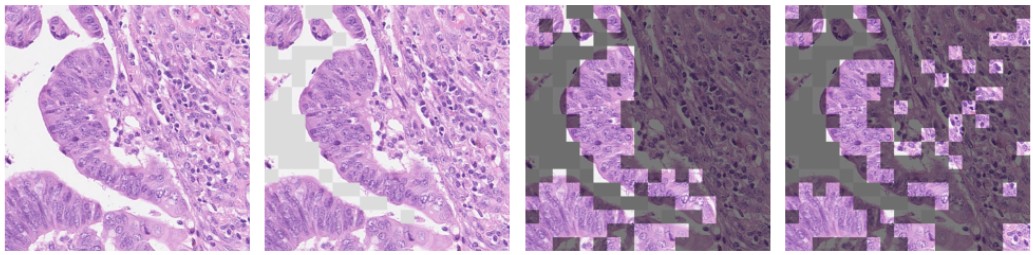

Figure A13: The interpretability output for a positive image in the CRC dataset. In this example, MILLI has identified some of the ground truth patches, but also additional patches that are not labelled as epithelial in the ground truth labelling. As the labelling was not exhaustive, these patches may contain epithelial nuclei without being labelled as such. By using MILLI, it is apparent that the model is using these patches in its decision-making, therefore further investigation into the types of nuclei in these patches would reveal more information about how the model makes its decisions; something that would not be possible without the interpretability output.

