# OpenReview forum: "Model Agnostic Interpretability for Multiple Instance Learning"
_ICLR.cc/2022/Conference — ICLR 2022 Poster_

### Official Review · Reviewer_xwk6 · 2021-10-29

**Correctness:** 2
**Technical Novelty And Significance:** 2
**Empirical Novelty And Significance:** 3
**Recommendation:** 5
**Confidence:** 5

**Main Review:**

On the positive side, the manuscript is well written and to a good extent its presentation and flow of contents is clear.

- The proposed method is simple and sufficient details are provided regarding its implementation and parameters. In this regard, I would not expect significant difficulties on the re-implementation of the proposed methods by third-parties.

- I appreciate the fact that the complexity of the proposed/discussed methods is provided. This is a detail that is sometime put aside.

- Finally, to the best of my knowledge, this might be one of the first works aiming at the task of explaining MIL methods.

On the negative side, I have the following concerns with the manuscript:

- While the manuscript covers an evaluation in multiple datasets, it would have been more insightful a more exhaustive analysis considering different MIL scenarios (e.g. different MIL assumptions, MIL regression vs. classification, other data modalities, etc.) see Wang et al., 2020 for reference.
A more comprehensive analysis would a provided deeper insight on the performance of the proposed method under different scenarios.

- When reporting results in Tables 1-3, more than one parameter is changed for the SHAP-based methods. In the manuscript, it is not properly motivated why this is the case.

- Reported results are not sufficiently conclusive, several trends can be observed in the results reported in Tables 1-3. More concretely, depending on the table different explanation methods lead the results.
Similarly, regarding the effect of sample size reported in Figure 3, it is stated that WeightedSHAP seems to be the most sample efficient of the SHAP variants while GuidedSHAP provides diminishing returns. However, these observations does not seem to hold in the other datasets as reported in Figures A13 and 14 from the appendix. Actually if you look at the plots from all the datasets together, it seems that GuidedSHAP provides the best tradeoff.

- The used performance metric seems to rely internally on the surrogate classes (supportive, neutral, refutive) that are also integrated as part of some of the proposed methods. Considering this observation makes me wonder whether the selected metric does work in favor of the proposed method.
In this regard, I would suggest to also report results using the perturbation-based method proposed by Samek et al., 2017 which can be applied without the assumption of such surrogate internal classes.

- Finally, when describing the proposed methods (e.g. Equation 1), it is stated that data is grouped per classes, however, it is not clear where this class information comes from. Could specify whether it comes from the ground-truth class annotations or the predicted classes?

References
- Wojciech Samek  Alexander Binder, Grégoire Montavon, Sebastian Bach, and Klaus-Robert Müller
Evaluating the visualization of what a Deep Neural Network has learned. TNNLS 2017

- Kaili Wang, Jose Oramas, and Tinne Tuytelaars, In Defense of LSTMs for addressing Multiple Instance Learning Problems. ACCV 2020.

**Summary Of The Paper:**

The manuscript aims at the design of a model-agnostic method of the the interpretability of models and methods addressing multiple instance learning (MIL) problem.
Towards this goal, six explanation methods are proposed, three of which are based on kernel-SHAP.
The proposed method is validated on the SIVAL, ColoRectal Cancer and a variant of the MNIST-Bags dataset.

**Summary Of The Review:**

As stated on my review, there are quite some merits regarding the presentation/reproducibility of the manuscript, and the simplicity of the proposed method.
However, I have concerns regarding the validation of the proposed method. On the one hand, it seems to cover a reduced set of MIL scenarios. Therefore, its generality cannot be guaranteed. In addition, at this point, it is hard to assess how conclusive the reported results really are. The fact that some of the proposed methods perform better in one or another dataset, suggests that the overarching goal of the manuscript -  of proposing an explanation method - has not been achieved.

---

> ### Author Response · Authors · 2021-11-15
> **Author Response to Reviewer xwk6**
>
> We provide the following responses to each of the outlined weaknesses:
>
> **Consider different MIL scenarios**
> In our work, we chose only to focus on classification tasks, however our evaluation does cover three different MIL scenarios: multi-class MIL with independent instances (SIVAL), multi-class MIL with instance interactions (4-MNIST-Bags), and SMIL with high witness rate and many instances per bag (CRC).  Due to the limited length of the paper, our ability to include a further dataset is restricted, especially as we are comparing several methods across several different MIL models. However, an expanded version of this work could indeed make use of a different version of 4-MNIST-Bags, potentially a regression task similar to the counting task in [1] . Furthermore, an expanded study could also include the LSTM model proposed by [1] as an additional MIL model.
>
> **Hyperparameter choices for Tables 1-3**
> In our evaluation, we tuned the number of samples and the value of $\alpha$ for each dataset. This is a realistic representation of how the methods would be used in practice: both variables are dataset dependent. We will make it clearer in our revised version of the paper where these hyperparameters came from.
>
> **Reported results are not sufficiently conclusive**
> Is it expected that different explanation methods perform better for different datasets. As discussed at the end of Section 5, the different methods have their own advantages. However, as part of our revisions, we are planning to incorporate further methods: LIME (please see our response to reviewer smzi), and also our own, MIL-specific method (please see our response to reviewer F67D), would should bring more conclusive results. While the efficacy of the methods may change across datasets, they are relatively consistent across models, i.e., a method that works well for one model on a given dataset will likely work well for all models on that dataset. Along with these new results, we will also re-run the sample efficiency experiments. In order to provide a better comparison across, we will bring all of the sample size plots into the main body of the text.
>
> **Additional metric**
> The NDCG@n metric does indeed rely on having (at least some) instance labels in order to generate a ground truth ordering. However, we explicitly do not use the instance labels in the training of the underlying MIL models, nor in our interpretability methods. The instance labels are strictly used only for evaluation, therefore there is no link between the selected metric and the methods we propose, i.e., the selected metric was not chosen to work in favour of the proposed methods. However, your suggestion of using the perturbation-based method from [2] is a really interesting proposal. The use of NDCG@n restricts the datasets we can use to those that have instance labels, so the obvious advantage of the perturbation-based method is that it does not require instance labels, and thus means these methods can be evaluated on more datasets. To this end, we have gone ahead and implemented the area over the perturbation curve metric (AOPC-R) to provide additional results for evaluating our methods. The trends are similar to what we observed when using NDCG@n. The obvious downside of using AOPC-R is that it is very expensive to compute, requiring many model passes to evaluate even a single bag. Therefore, in our revised version of the paper, we will include the AOPC-R results, but only for the SIVAL and 4-MNIST-Bags datasets, as it is too expensive to compute for the CRC dataset. NDCG@n requires instance labels but is quick to compute; AOPC-R does not require instance labels but is expensive to compute.
>
> **Class information in methods**
> We are unsure what you mean by the statement: "it is stated that data is grouped per classes". If, for Equation 1, you are referring to $F_c$, this is the output of the MIL model $F$ for a particular class $c$. For example, in the 4-MNIST-Bags dataset, there are four possible classes. For an input bag $X$, the output $F(X)$ is a vector of length four, and $F_c(X)$ is simply the $c^{th}$ entry in this vector. So we are not grouping the data by class, we are merely observing the predictions of the data with respect to a particular class. The proposed methods produce instance importances for each instance with respect to each class, i.e., we enumerate $c$ over all the possible classes $c \in C$. We are happy to change the relevant wording in the revised version of the paper to aid with clarity.
>
> [1] KailiWang, Jose Oramas, and Tinne Tuytelaars. In defense of lstms for addressing multiple instance learning problems. In Proceedings of the Asian Conference on Computer Vision, 2020.
> [2] Wojciech Samek, Alexander Binder, Gr´egoire Montavon, Sebastian Lapuschkin, and Klaus-Robert M¨uller. Evaluating the visualization of what a deep neural network has learned. IEEE transactions on neural networks and learning systems, 28(11):2660–2673, 2016.

---

> > ### Comment · Reviewer_xwk6 · 2021-11-19
> > **Re: Author Response to Reviewer xwk6**
> >
> > Thanks for your feedback and for addressing my initial review,
> >
> > I do  understand the challenges regarding evaluating a larger variety of MIL scenarios given the page limit. But I am always of the idea that this type of analysis can always help paint a better picture regarding the strengths and weaknesses of a given method.
> >
> > Your feedback regarding the Hyperparameters choices for Tables 1-3 and the class information in methods did clarify my doubts. I would indeed suggest to revise those aspects in the paper to ensure clarity.
> >
> > Thanks to for the additional results regarding the AOPC-R metric. Having an additional metric can further confirm the observations that you were making with the metric that you initially adopted.
> >
> > Regarding the "inconclusiveness" of the reported results. It is indeed normal that different explanation methods have variable performance on different datasets. However, I would expect overall trends, e.g. method-1 is in the majority of the datasets better than method-2, to be consistent. If not this suggest that the methods in question are tailored to the problems in the considered datasets. If performance of a method shows high variation when changing datasets, perhaps it is not method that can generalize to other problems and it should be already a red flag if high variation is observed already in a small set of datasets.

---

> > > ### Author Response · Authors · 2021-11-19
> > > **Continued Author Response to Reviewer xwk6**
> > >
> > > Thank you for your further comments and confirming that our revisions resolved your concerns regarding a) hyperparameter choices, b) class information, and c) additional metrics.
> > >
> > > **Inconclusiveness of the reported results**
> > > We have improved our WeightedSHAP algorithm to be able to adapt better to different datasets, improving its generalisability. It now provides more consistent performance across all datasets. Our revised version of this paper will contain this new improvement, with an evaluation including AOPC and more datasets (MUSK, TIGER, ELEPHANT, and FOX).
> > >
> > > **Investigation of different MIL scenarios**
> > > While the focus of this paper is on classification tasks, we look forward to conducting a thorough analysis of additional MIL scenarios (including regression) in future research. In our revised version of this paper, we will mention the tasks in [1] as an area of future work.
> > >
> > > [1] KailiWang, Jose Oramas, and Tinne Tuytelaars. In defense of lstms for addressing multiple instance learning problems. In Proceedings of the Asian Conference on Computer Vision, 2020.

---

### Official Review · Reviewer_F67D · 2021-11-02

**Correctness:** 3
**Technical Novelty And Significance:** 3
**Empirical Novelty And Significance:** 3
**Recommendation:** 5
**Confidence:** 4

**Main Review:**

The paper presents a suite of sampling-based approaches to compute Shapley values for interpretable multiple instance learning. The focus of the paper is on answering "which" questions (for identifying key instances in a bag) and answering "what" questions (for identifying which positive classes are supported by key instances).

The motivation for this paper is solid as there is not much work in interpretable MIL, especially with more than one positive class. The algorithms make technical sense and I can see them being useful in real-world applications such as colon cancer classification.

Unfortunately, the paper suffers from a number of weaknesses. The main problem with the paper is its lack of novelty. The instance attribution ideas in Section 3.2 are fairly obvious things to do. Secondly, the key ideas for dealing with instance interactions (e.g. Shapley values for explainability, sampling for Shapley value computation) come from existing work and the authors are performing minor tweaks to them through their weighted sampling technique.

Furthermore, the evaluation could be much stronger. There are only three datasets used in the evaluation section, with two of them having independent instances. It would help to have more real-world datasets involving instance interactions. The authors use the 4-MNIST-Bags dataset as an example of a dataset with instance interactions. The positive classes are artificially generated, which is acceptable for evaluation, but the authors could produce many more variants of the 4-MNIST-Bags for a more thorough evaluation.

There are two terms used in this paper that are used somewhat differently in the context of the wider machine learning literature and this use is confusing. Explicitly clarifying these terms would greatly improve the text.

The first is "interpretability". From the text, I think the authors intend interpretability to be the correct identification of which are the key instances as well as the correct identification of what classes the key instances support. This use of the term interpretability different from what is commonly referred to as interpretability in the XAI literature, in which interpretability captures how well a human user can understand why a machine learing algorithm makes a particular prediction. This latter definition of interpretability is a human-centric concept and thus requires a user study to measure it; it is not captured by the NDCG@N metric.

The second confusing term is "interaction between instances". I think the authors are referring to the fact that multiple instances can support different classes. However, other work in the machine learning literature (e.g. Adams and Marlin (2017), Guan et al. (2016)) consider "interactions" to be relationships between the instances e.g. the instances are intervals of a time series and are thus correlated with each other.

This second point of confusion makes it unclear how the sampling approaches in Section 3.3 capture interactions between instances. The random sampling treats each member of the coalition to be independently drawn. The guided sampling adds a weight to the random sampling to provide a bias towards small coalitions, but this weight still treats each member independently. Finally, the weighted sampling relies on independent instance methods to rank. This section of the paper could be improved if the authors could explicitly clarify how the sampling approaches moves away from the independence of instances and capture interactions.

References

Adams, R. J. and Marlin, B. M. (2017). Learning Time Series Detection Models from Temporally Imprecise Labels. In Proceedings of the 20th International Conference on Artificial Intelligence and Statistics.

Guan, X., Raich, R. and Wong, W-K. (2016). Efficient Multi-Instance Learning for Activity Recognition from Time Series Data Using an Auto-Regressive Hidden Markov Model. In Proceedings of the 33rd International Conference on Machine Learning.


**Summary Of The Paper:**

This paper presents model-agnostic Shapley value approaches for interpretable multiple instance learning, with a focus on identifying key instances and identifying which positive classes these key instances support.

**Summary Of The Review:**

The paper addresses the interpretability of multiple instance learning, which could use more attention in the literature, but it has weaknesses in terms of the novelty of ideas, the clarity of terms and the experimental evaluation.

---

> ### Author Response · Authors · 2021-11-15
> **Author Response to Reviewer F67D**
>
> We provide the following responses to each of the outlined weaknesses:
>
> **Lack of novelty**
> While our instance attribution methods in Section 3.2 may appear obvious, no other work to our knowledge has proposed such methods for interpretability in MIL (and they are effective when the instances are independent). For the use of the SHAP, while we are adapting existing methods for MIL, we are again still the first to implement and show that these methods work for MIL. However, we do concede that the development of the weighted sampling technique for SHAP only represents a minor increment over existing methods. To increase the novelty of our work, we are currently working on developing a more sophisticated MIL-specific interpretability method as a stand-alone alternative to SHAP (and LIME, please see our response to reviewer smzi). This new approach utilises the same ideas as used in our weighted sampling approach, but is its own unique method. We will include this method in our revised version of the paper.
>
> **Additional datasets**
> While more datasets may further strengthen our evaluation, we specifically chose these three datasets to analyse specific aspects of MIL datasets: SIVAL as feature-based, classical MIL dataset, 4-MNIST-Bags as a dataset with instance interactions, and CRC as a larger dataset (greater number of instances per bag and with a higher witness rate) to the test the scalability of our methods. Due to the limited length of the paper, the scope to include a further dataset is restricted, especially as we are comparing several methods across several different MIL models. However, in an expanded version of this work, further datasets, including additional variants of the 4-MNIST-Bags datasets, should indeed be used. Please see our response to Reviewer xwk6, in which we discuss using 4-MNIST-Bags for a MIL regression problem.
>
> **Use of "interpretability"**
> We do indeed use interpretability to mean "the correct identification of which are the key instances as well as the correct identification of what classes the key instances support". However, in a similar vein to our response to reviewer Mxks, we explicitly chose to use "interpretability'' rather than "explainability'', as we feel explainability refers to explaining things in more human terms, as opposed to technical interpretations of model decision-making. Under this definition of interpretability as having a technical focus, NDCG@n is an appropriate measure. However, if the focus was on explainability, i.e., human understanding of decision-making, then we agree, a user study would be required. We are happy to clarify what we mean by interpretability in our revised version of the paper.
>
> **Use of "interaction between instances"**
> We use interaction between instances to refer to cases when the co-occurrence of two (or more) instances in the bag changes the bag label (please see our discussion of this in our response to reviewer smzi). This implies a relationship between the two (or more) types of instances, as they have different meanings depending on the context of the bag, i.e., on their own they mean something different to what they mean when observed together. Again, we are happy to clarify what we mean by the interactions between instances in our revised version of the paper.
>
> **Unclear how the sampling approaches capture interactions between instances**
> The sampling processes themselves are not responsible for capturing the instance interactions. Rather, it is the use of a surrogate local model (the weighted linear model $g$) that is locally faithful to the MIL model $F$ that captures the instance interactions. However, the efficacy of the surrogate local model, both in how faithful it is to the original MIL model and how well it captures the instance interactions, is dependent on the way the coalitions are sampled and weighted. To this end, our weighted sampling approach is more likely to sample coalitions that enable the surrogate local model to better capture the instance interactions (as demonstrated through strong performance on the 4-MNIST-Bags dataset). We agree to make this much clearer in our revised version of the paper.

---

> > ### Comment · Reviewer_F67D · 2021-11-30
> > **Thank you for the revisions**
> >
> > The authors have improved the paper with their latest revisions. I like the direction they have gone with the MILLI algorithm and I agree that a ranking-based approach is a promising direction for this line of work. I bumped my score a little bit but I still feel that the paper needs a bit more work for acceptance. Despite some nice aspects, the MILLI algorithm has its drawbacks -- 1) it is not clear how to tune alpha and beta hyperparameters when one applies the algorithm out-of-the-box to a new problem and 2) it is quite computationally expensive to obtain the ranking. Even with the revisions, I stilI feel that the overall contribution is somewhat minor. With more work, perhaps by following up on the future directions mentioned at the end of the revised version, the authors will have sufficient technical contributions for a solid accept.

---

> > > ### Author Response · Authors · 2021-11-30
> > > **Continued Author Response to Reviewer F67D**
> > >
> > > Thank you for your further comments and for having a look at our updated version of the paper.
> > >
> > > Regarding the drawbacks of MILLI:
> > >
> > > **1) Tuning of alpha and beta.** We discuss how we tuned the hyperparameters for our interpretability methods in Appendix A.5. As described there, we tuned alpha and beta for each method using a grid search, and found that the discovered hyperparameters match our expectations with regards to the sampled coalition size for MILLI (e.g., for the CRC, Musk and TEF datasets, the sampling is heavily biased towards smaller coalitions). A similar approach to tuning could be used for applying MILLI to an out-of-the-box algorithm and different dataset.
> > >
> > > **2) Quite computationally expensive to obtain the ranking.** We are unsure which ranking the reviewer is referring to here. If this is in regards to the overall computational complexity of MILLI, then we would argue that, due to our sampling approach, it is less expensive to compute than the comparable methods of LIME and SHAP as, on average, it is more sample efficient (see Figure 4).
> > >
> > > We would also like to restate the main contributions of our paper, as we feel they are sufficiently novel for acceptance:
> > >
> > > 1) This is the first study (to the best of the authors' knowledge) to investigate the problem of model-agnostic interpretability for MIL.
> > > 2) We present three novel approaches for MIL model-agnostic interpretability under the assumption that the instances are independent.
> > > 3) We then relax this assumption and present a novel local surrogate post-hoc interpretability method, specially designed for MIL, that is able to deal with instance interactions, and can be tuned for different types of MIL classification problems.
> > > 4) We are the first to conduct a study comparing existing inherently interpretable MIL models with a) our new methods, and b) out-of-the-box interpretability such as LIME and SHAP. Further note that LIME and SHAP had not been discussed in relation to MIL in any prior works.
> > >
> > > We would like to once again thank you for your comments.

---

### Official Review · Reviewer_MxkS · 2021-11-03

**Correctness:** 4
**Technical Novelty And Significance:** 2
**Empirical Novelty And Significance:** 2
**Recommendation:** 5
**Confidence:** 2

**Main Review:**

The paper is nicely written and easy to follow, but, in my opinion, it totally overlooks existing prior work that should be cited. In particular, I would like to see comparisons with the techniques presented in the following paper: https://www.jmlr.org/papers/volume21/18-811/18-811.pdf It is from 2020 and already mentioned interpretability, and it should be used as a baseline.

The reason mentioned above is the strongest one to ground my rejection score for this paper. Another one is that SHAP looks like an arbitrary choice from the paper. There are many explanation mechanisms that can be used, and it is not clear why SHAP is the only one you try (besides the simple three methods you introduce at the beginning)

Apart from those negative comments, the paper is well written and easy to read. Only one comment regarding the terminology you use in your paper.  The term "interpretable" usually refers to the fact that you can look at the model and understand how predictions are computed. Yours is more an "explainability" method.

**Summary Of The Paper:**

The paper presents six methods for explaining the output of MIL models. The idea is pretty simple but, indeed, effective. The most advanced method is based on SHAP, and it is used to assign to each instance in the bag a value that weights its importance.
Experiments confirm that the method is able to improve over competitive baselines.

**Summary Of The Review:**

The paper presents a SHAP-based explainability method for MIL models. The paper is easy to read, but it lacks the sufficient comparison with proper baselines. I mentioned a paper that should be used to build a stronger baseline.

---

> ### Author Response · Authors · 2021-11-15
> **Author Response to Reviewer MxkS**
>
> We provide the following responses to each of the outlined weaknesses:
>
> **Comparison to prior work**
> The paper you suggest [1], focuses on multi-multi-instance learning (MMIL), where the data is represented as bags within bags. This data representation means the models and interpretability techniques outlined are not applicable to the general MIL paradigm which we discuss in our work, i.e., the interpretability approach outlined in the suggested paper is dependent on the use of MMIL data and models, therefore is not an applicable baseline for our work.
>
> **Arbitrary use of SHAP**
> We agree that the use of SHAP is a somewhat arbitrary choice of method when there are alternatives available. In response, we direct you to our response to Reviewer smzi, in which we detail our initial results when using LIME instead of SHAP. We are also in the process of developing our own MIL-specific method as a further alternative to SHAP (please see our response to Reviewer F67D). We aim to incorporate both LIME and our new method in the revised version of the paper.
>
> **Use of "interpretability" rather than "explainability"**
> While these two terms are often used interchangeably in the literature, for consistency we decided to stick to only using one term. Our argument for using "interpretability" rather than "explainability" is that the outputs of our methods are still quite technical: they're numeric and tied to the model/classes etc. We feel "explainability" should be reserved for explaining things in more human terms, for example textual outputs using simple wording, i.e., outputs that do not detail the technical aspect of the decision-making process.
>
> **References**
> [1] Alessandro Tibo, Manfred Jaeger, and Paolo Frasconi. Learning and interpreting multi-multi-instance learning networks. J. Mach. Learn. Res., 21:193–1, 2020.

---

> > ### Comment · Reviewer_MxkS · 2021-11-17
> > **Still not convinced**
> >
> > Thanks for your comments.
> >
> > As I stated in my original review, the relationships with existing related work are still unclear. As other reviewers have mentioned, some datasets are classic for MIL but that you have overlooked. Also, the result of Tibo et al. it's just a starting point, and many follow-up papers have not been mentioned in the paper itself.
> >
> > I would stick to my original score.

---

> > > ### Author Response · Authors · 2021-11-17
> > > **Continued Author Response to Reviewer MxkS**
> > >
> > > Thank you for your further comments.
> > >
> > > With regards to the related work on MMIL, we provide the following statement, which we are happy to include in Section 2 (Background and Related Work) of our paper:
> > >
> > > *A related piece of work on interpretability in MIL is Tibo et al. (2020) [1], which considers interpretability within the scope multi-multi-instance learning (MMIL; Tibo et al. (2017) [2]; Fuster et al. (2021) [3]). In MMIL, the instances within a bag are arranged into into further bags, giving a hierarchical bags-of-bags structure. This extension of MIL is naturally applicable to certain types of datasets, however the interpretability techniques presented by Tibo et al. (2020) [1] are model-specific as they are only designed for MMIL networks and MMIL datasets. In this work, we focus on traditional MIL, (i.e., without the nested bag structure; only using 'flat' bag representations of instances) as it is a much more widely used and well studied learning paradigm. However, a future extension to this work could be to apply our techniques in a MMIL setting and compare them to existing MMIL-specific techniques.*
> > >
> > > With regards to overlooking classical MIL datasets, as we state in our response to Reviewer smzi, many of the classical MIL datasets do not have instance labels, therefore it is difficult to evaluate the different interpretability methods on them as we have no ground truth instance ordering to compare to. We specifically chose the SIVAL dataset as it is one of the only classical MIL datasets that has instance labels. However, for further experiments on classical datasets, we can use the AOPC metric as suggested by Reviewer xwk6 to evaluate the efficacy of the different interpretability methods without needing instance labels. AOPC does not scale well with larger datasets (a greater number of instances per bag means the evaluation is very expensive to compute), however for classical MIL datasets such as MUSK1, MUSK2, FOX, TIGER, ELEPHANT (as mentioned by Reviewer smzi), AOPC could be applied. We are currently in the process of applying AOPC to the SIVAL and 4-MNIST-Bags datasets, and will also apply AOPC to additional classic MIL datasets. We will add the AOPC results for the benchmark dataset to our revised version of the paper.
> > >
> > > [1] Alessandro Tibo, Manfred Jaeger, and Paolo Frasconi. Learning and interpreting multi-multi-instance learning networks. J. Mach. Learn. Res., 21:193–1, 2020.
> > > [2] Alessandro Tibo, Paolo Frasconi, and Manfred Jaeger. A network architecture for multi-multi-instance learning. In Joint European Conference on Machine Learning and Knowledge Discovery in Databases, pages 737–752. Springer, 2017.
> > > [3] Saul Fuster, Trygve Eftestøl, and Kjersti Engan. Nested multiple instance learning with attention mechanisms. arXiv preprint arXiv:2111.00947, 2021

---

### Official Review · Reviewer_smzi · 2021-11-03

**Correctness:** 3
**Technical Novelty And Significance:** 3
**Empirical Novelty And Significance:** 2
**Recommendation:** 6
**Confidence:** 2

**Main Review:**

Strengths
1. This paper formulates the interpretability requirements. Important points are 1) defining the conditional contribution for a particular class ; 2) allowing the contribution to be negative, which indicates that the instance refutes some class.
2. Several straightforward methods and Shapely value based methods are evaluated on three datasets.
3. Authors propose a novel weight-sampling strategy to improve sampling in SHAP.
4. The paper is well written and easy to read.

Weaknesses
1. I'm not sure whether the interpretability requirements for MIL are original ideas, since MIL is not a new problem. Moreover, these requirements have already been defined and addressed in studies of feature importance for standard single instance models. Computing instance importance and feature importance are similar problems.
2. Related methods such as LIME[1] should be discussed. LIME is also a model-agnostic local interpretability method. It should be naturally suitable for interpreting MIL models.
3. Classical MIL datasets (MUSK1, MUSK2, FOX, TIGER, ELEPHANT) used in [2,3] are missed in the experiments. Can authors explain why?
4. In the first part of Section 5, it's not clear why the demonstrated case shows interactions between instance 8 and 9? And what's the evidence that WeightedSHAP succeeds in taking account of the interaction?

[1] Marco Tulio Ribeiro, Sameer Singh, Carlos Guestrin. “Why Should I Trust You?": Explaining the Predictions of Any Classifier.

[2] Xinggang Wang, Yongluan Yan, Peng Tang, Xiang Bai, and Wenyu Liu. Revisiting multiple instance neural networks.

[3] Maximilian Ilse, Jakub Tomczak, and Max Welling. Attention-based deep multiple instance learning.

**Summary Of The Paper:**

This paper introduces approaches to interpret Multiple Instance Learning (MIL) models. These approaches are designed to quantify the contribution of each instance given a specific class. Authors provide empirical studies about these methods. The single instance evaluation and leave one out method show good performance when instances are independent with each other. The Shapely value based methods are much better when instances have interactions. All these methods exhibit higher interpretability than the inherent interpretations provided by the selected MIL training methods.


**Summary Of The Review:**

I recommend weak rejection for now, considering the weaknesses listed above.

---

> ### Author Response · Authors · 2021-11-15
> **Author Response to Reviewer smzi**
>
> We provide the following responses to each of the outlined weaknesses:
>
> **Originality of MIL interpretability requirements**
> The originality in the interpretability requirements for MIL lies in the definition of the *which* and the *what* questions. As we discuss in Section 2, prior works that propose interpretable MIL models (such as attention pooling [2] and graph neural networks [3]) do not provide class-specific interpretations, nor are the interpretations able to refute certain outcomes. Without have a defined set of requirements that MIL interpretability methods should achieve, it is not possible to compare the merits of different methods. While the requirements we propose are related to existing ideas for single instance models (such as LIME and SHAP), it is still necessary to explicitly define them for MIL, which is what we achieve in Section 3.1 of our work.
>
> **Comparison to LIME**
> We are in the process of running further experiments where we use the LIME weight kernel in place of the SHAP kernel. Our initial results on the MNIST dataset show that the LIME kernel actually marginally outperforms the SHAP kernel: WeightedLIME using L2 distance gives an NDCG@n of 0.9319 compared to WeightedSHAP's 0.9254. Once we have expanded these experiments to the other datasets, we will update the paper to include the new results using the LIME kernel.
>
> **Use of existing benchmark datasets**
> In order to evaluate the efficacy of the produced interpretations, we require ground truth instance labels. Without instance labels, it cannot be determined whether the interpretations for each instance are correct with respect to each class. Therefore, we are limited to only using datasets with instance labels. Unfortunately, many of the classical MIL datasets do not have instance labels. Therefore, we cannot use them. We note this in Section 4.1 of our work, and also direct you to Table 3 of [1], which provides details on the properties of benchmark MIL datasets. The SIVAL dataset is only one of three listed datasets the has instance labels (the other two being Birds and Newsgroups).
>
> **Instance interaction**
> For the 4-MNIST-Bags dataset, the interaction between instances is the co-occurrence of the **8** and **9**. When observed on their own, an **8** and a **9** represent class one and class two respectively, but when observed together, they represent class three, i.e., their co-occurrence changes the bag label. Therefore, the occurrence of an **8** supports classes 1 and 3 but refutes class 2, as, by definition of the classes, **8s** can never been in class 2 bags. Similarly, the occurrence of an **9** supports classes 2 and 3 but refutes class 1, as, by definition of the classes, **9s** can never been in class 1 bags. The example given in Section 5 (Figure 1) shows that the WeightedSHAP interpretation uncovers this same relationship. This demonstrates that WeightedSHAP identifies the co-occurrence of the **8** and **9** as an indicator of class 3, something that is not possible when only observing each instance independently.
>
> **References**
> [1] Marc-Andre Carbonneau, Veronika Cheplygina, Eric Granger, and Ghyslain Gagnon. Multiple instance learning: A survey of problem characteristics and applications. Pattern Recognition, 77:329–353, 2018.
> [2] Maximilian Ilse, Jakub Tomczak, and Max Welling. Attention-based deep multiple instance learning. In International conference on machine learning, pages 2127–2136. PMLR, 2018.
> [3] Ming Tu, Jing Huang, Xiaodong He, and Bowen Zhou. Multiple instance learning with graph neural networks. arXiv preprint arXiv:1906.04881, 2019.

---

> > ### Comment · Reviewer_smzi · 2021-12-02
> > **Response to authors**
> >
> > As authors’ response solved most of my questions, I would like to increase my score to 6. This paper makes a first step to study model-agnostic interpretability in MIL tasks. This should be of interest in many real world applications. However, this paper is mostly an empirical study without surprising findings, and I still think the MIL interpretability definition is not an important contribution as it has been already established in standard ML tasks. For these reasons, I would only recommend a weak acceptance.

---

> ### Comment · Area_Chair_NAJb · 2021-11-30
> **Response to author rebuttal?**
>
> Hi Reviewer smzi! Could you update your review based on author's comments? They provided preliminary additional results (e.g., LIME baseline) based on your comments.

---

### Author Response · Authors · 2021-11-20
**Preliminary Revised Submission**

Thank you to all reviewers for their initial comments and ongoing discussion about our work.
We have uploaded a preliminary revised version of our paper based on reviewer feedback.

The notable changes are as follows:

**Improved version of WeightedSHAP (Section 3.3)**
We have replaced WeightedSHAP with a new MIL-specific local surrogate method for MIL interpretability (MILLI). This new method overcomes the previous issues with WeightedSHAP, and is a better performing and more novel approach.

**Inclusion of LIME (Section 4.3, Section 4.4, Appendix A.5)**
Based on comments from Reviewer smzi, we have included an analysis against LIME as well as SHAP.

**Additional evaluation metric  (Section 4.1, Appendix A.3, Appendix A.4)**
Based on comments from Reviewer xwk6, we have added results using an further metric (AOPC). We have re-run our previous experiments to give updated results.

**Additional datasets (Appendix A.4)**
Given that we are now also using AOPC as an evaluation metric, we can now run experiments on datasets that do not have instance labels (previously we were restricted to only using datasets with instance labels). In light of this, and as the request of Reviewers smzi, MxkS and F67D, we have included additional results on the classical MIL datasets Musk, Tiger, Elephant and Fox.

We have also addressed other minor issues from reviewers, such as detailing hyperparameters (Appendix A.5), improving our discussion of prior work (Section 2), clarifying the use of "interpretability" (Section 2), and providing more conclusive results (Section 4.4, Section 5).

We welcome any further comments on the revised version of the paper. Note that we will also update our supplementary material in due course.

---

### Author Response · Authors · 2021-11-22
**Further Revised Submission**

Following our Preliminary Revised Submission on the November 20th, we have now submitted a further revision.

The changes for our new revision are detailed below. Please also see our previous comment regarding the changes for the Preliminary Revised Submission.

**Updated Results**
We have updated our results (Section 4.4) to now include the AOPC-R results for the SIVAL and 4-MNIST-Bags dataset. Previously, these results were in Appendix A.4. of the Preliminary Revised Submission. By adapting the tables in Section 4.4, we are able to move the AOPC-R results from the Appendix into the main body of the paper, allowing us to discuss the AOPC-R results alongside the NDCG results for the SIVAL and 4-MNIST-Bags dataset. The results themselves have not changed from the Preliminary Revised Submission, only their presentation.

We also made minor text changes (grammar, spelling etc.), as well as a minor change to the abstract (we previously stated we use three datasets, which, with the inclusion of the Musk, Tiger, Elephant, and Fox datasets, is no longer true). With this submission we also added our revised supplementary material, containing updated code, results, and outputs for the new experiments.

We would like to once again thank all the reviewers for the initial comments and ongoing discussion about our work. By following your suggestions, we feel we have greatly expanded upon the original version of our paper, both in our methodology and in our experiments. We welcome any further comments on the revised version of the paper, and are happy to discuss it further through to November 29th.

---

### Comment · Area_Chair_NAJb · 2021-11-24
**Discussion Period!**

Dear reviewers,

Authors have provided a detailed response as well as updated the manuscript.
Could you take a look and comment on how this changes your initial evaluation?
Even when your evaluation does not change, it would be helpful to know why.

best,
Area Chair

---

### Decision · Program_Chairs · 2022-01-20

**Decision:**

Accept (Poster)

**Comment:**

The paper studies interpretability in multi instance learning (where model is trained with a label provided for a bag of instances). The author proposes model-agnostic weight-sampling strategy to improve sampling in prior methods such as (SHAP), and evaluate their performance on three datasets (and authors provided results on more datasets during rebuttal).

All reviewers agree the paper is well written and well motivated. The paper presents a simple but meaningful extensions to existing interpretability study and will be helpful for the community. Reviewers had some concerns with the comprehensiveness of the evaluation, the strength of their proposed results, and the originality/novelty of the paper. The authors have provided further experimental results on new datasets as well as additional baselines. Given the study of MIL setting in interpretability is scarce, I am leaning towards the acceptance.